# CAMO: CATEGORY-AGNOSTIC 3D MOTION TRANSFER FROM MONOCULAR 2D VIDEOS

## ABSTRACT

Motion transfer from 2D videos to 3D assets is a challenging problem, due to inherent pose ambiguities and diverse object shapes, often requiring category-specific parametric templates. We propose CAMO, a category-agnostic framework that transfers motion to diverse target meshes directly from monocular 2D videos without relying on predefined templates or explicit 3D supervision. The core of CAMO is a morphology-parameterized articulated 3D Gaussian splatting model combined with dense semantic correspondences to jointly adapt shape and pose through optimization. This approach effectively alleviates shape-pose ambiguities, enabling visually faithful motion transfer for diverse categories. Experimental results demonstrate superior motion accuracy, efficiency, and visual coherence compared to existing methods, significantly advancing motion transfer in varied object categories and casual video scenarios.

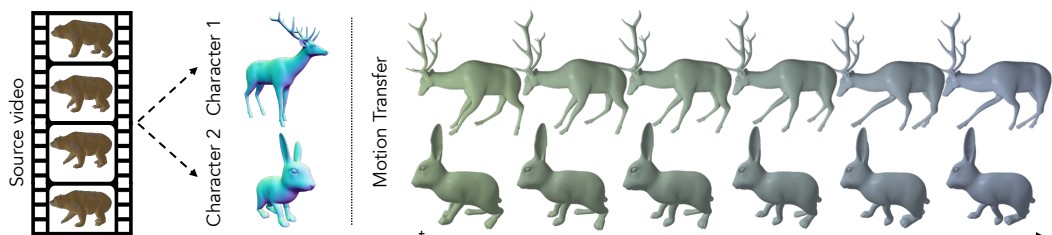

Figure 1: **Conceptual overview of CAMO.** Our method directly transfers articulated motion from 2D video to diverse target objects, without requiring 3D reconstruction of the source or any parametric templates.

## 1 INTRODUCTION

Efficient 3D character animation remains an important goal in both computer graphics research and content industries such as film (Bregler, 2007), interactive media (Rachmavita, 2020), and robotics (Arduengo et al., 2021). Motion transfer techniques (Aberman et al., 2020; Liao et al., 2022) provide an efficient alternative to manual keyframing or marker-based motion capture by enabling the reuse of existing animations across different characters.

However, a major limitation of many existing methods is their reliance on precomputed 3D sequences, such as articulated skeletons (Aberman et al., 2020) or sparse 3D keypoints (Chen et al., 2023). Acquiring such high-fidelity 3D data is often expensive or impractical in real-world scenarios. To address this data scarcity, recent works (Wang et al., 2023; Muralikrishnan et al., 2024) have explored extracting motion cues directly from readily accessible 2D monocular videos. A common strategy within this domain involves a two-stage reconstruct-then-retarget approach. In this process, a 3D proxy representation of the source subject is first reconstructed from the 2D video, and this intermediate representation is then fed into established 3D-to-3D motion transfer techniques.

Despite demonstrating effective retargeting performance under controlled conditions, these sequential pipelines inherently possess several limitations. A primary limitation stems from their dependence on category-specific priors, such as parametric template models (Loper et al., 2015; Zuffi et al., 2017), which require large-scale, high-fidelity training data. Although models built on such priors (Kanazawa et al., 2018; Zhang et al., 2021; Rueegg et al., 2022) achieve robust and transferable pose estimation within the structural biases of their target domains, their ability to generalize

to diverse shapes and semantic categories remains limited. Furthermore, the cascaded structure of these pipelines can lead to error propagation, where inaccuracies from the reconstruction stage detrimentally impact the fidelity of the final transferred motion.

Our category-agnostic motion transfer framework, **CAMO**, adopts an alternative strategy to conventional reconstruct-then-retarget pipelines. Rather than relying on intermediate 3D reconstructions of the source, we directly project the target character into the 2D observation space, enabling pose optimization purely through image-space supervision. Specifically, we repurpose articulated 3D Gaussian splatting (Yao et al., 2025) (articulated-GS), originally developed for reconstructing articulated animatable objects from 2D videos, to facilitate motion transfer.

CAMO extends this by explicitly modeling morphological differences between source and target characters. Structural variations are decomposed from the target's original shape and adapted to transfer the source motion while preserving topology. To complement this morphology-adaptive optimization and further mitigate shape-pose ambiguity, dense semantic correspondences are established between the 2D source frames and the 3D target mesh, providing semantic guidance for coherent pose recovery. This integration of structural modeling and semantic correspondence guides both visually plausible and semantically coherent pose optimization processes, enabling robust generalization across diverse categories and complex motions. Fig. 1 illustrates the overview of CAMO.

We comprehensively validate CAMO on synthetic benchmarks spanning diverse categories such as humanoids, quadrupeds, and other non-standard animals, as well as on real-world monocular videos. Across all these settings, CAMO consistently preserves motion fidelity and generalizes across diverse morphologies, achieving substantial improvements in both PMD ($\downarrow$) and FID ($\downarrow$), with reductions reaching up to 85% on the challenging categories compared to state-of-the-art methods.

## 2 RELATED WORK

**Motion transfer between 3D assets.** Traditional techniques in motion transfer have leveraged 3D skeletal structures to enable efficient retargeting across various characters (Gleicher, 1998; Villegas et al., 2018; Aberman et al., 2020; Villegas et al., 2021; Chen et al., 2023). These approaches commonly build upon category-specific skeletal priors, which enable effective performance within their target domains but constrain their generalization to categories outside those domains.

Beyond skeleton-based approaches, skeleton-free deformation methods (Gao et al., 2018; Wang et al., 2020; Liao et al., 2022; Wang et al., 2023; Muralikrishnan et al., 2024; Yoo et al., 2024) are independent from explicit skeletal models, relaxing categorical constraints. Nevertheless, these approaches typically rely on high-quality 3D motion data, which is generally not available for objects across diverse categories. As a result, generalizing these methods to a wider variety of object categories remains a notable challenge, primarily due to the substantial cost and scarcity of such 3D data.

**Shape and pose estimation from 2D videos.** Another line of research focuses on capturing 3D pose from monocular video. These methods achieve impressive reconstructions within specific domains, often leveraging parametric templates. Representative works include human pose estimation (Zhang et al., 2021; Goel et al., 2023) with SMPL (Loper et al., 2015), and quadruped pose estimation (Rüegg et al., 2023; Lyu et al., 2024) with SMAL (Zuffi et al., 2017). Although effective in domains with abundant 3D scan data, these methods are constrained by their reliance on parametric templates, which limits generalization to categories without extensive 3D pose annotations.

Recent approaches (Yao et al., 2022; Wu et al., 2023a;b; Aygun & Mac Aodha, 2024; Li et al., 2024) explore parametric template-free construction of articulated models from image collections. While promising for intra-class generalization without strong parametric template priors, these methods often struggle to generalize across categories. Uzolas et al. (2023) and Yao et al. (2025) inherently avoid this limitation by employing per-scene optimization to directly decompose shape and skeletal pose from individual dynamic scene observations. However, as their focus lies in reconstruction, their ability to retarget motion to novel characters remains underexplored.

Specifically targeting character animation, auto-rigging methods (Song et al., 2025a; Zhang et al., 2025a) predict the skeleton and skinning weights of a 3D asset to apply motion extracted from videos or reconstructed mesh sequences. However, these methods typically require a complete mor-

phological (Song et al., 2025a) or skeletal structural correspondence (Zhang et al., 2025a) between the motion source and the target 3D character.

**2D to 3D motion transfer.** Existing 3D-to-3D motion transfer frameworks (Wang et al., 2023; Muralikrishnan et al., 2024) extend to the 2D domain by combining parametric template-based pose and shape estimators (Zhang et al., 2021; Rueegg et al., 2022) with 3D pose transfer techniques. These shape estimators are typically demonstrated on humanoid or quadruped characters respectively, where the reliance on categorical templates (Loper et al., 2015; Zuffi et al., 2017) fundamentally limits their ability to generalize to novel categories. Moreover, we observe that sequentially combining independently trained components often leads to cumulative errors, ultimately degrading the fidelity of transferred motion.

Maheshwari et al. (2023) propose a category-agnostic approach that removes template priors, transferring motion from RGB-D videos to 3D meshes by estimating skeletal motion from reconstructed meshes; its performance, however, hinges on accurate depth input, limiting robustness in casual or monocular RGB settings. In contrast, Fu et al. (2024) and Zhang et al. (2024a) achieve 2D-to-3D motion transfer without depth by reconstructing motion with neural bones (Yang et al., 2022) or by leveraging image-to-3D generative models (Liu et al., 2023). Despite improved generalizability, these approaches remain tied to intermediate reconstruction stages (e.g., pseudo-3D supervision or skeletonization), which makes them sensitive to reconstruction errors and less robust under large morphological variations.

In contrast, we directly leverage 2D RGB videos as motion sources through morphology-adaptive shape and pose parameter optimization. By bypassing intermediate 3D reconstruction, our approach mitigates reconstruction errors and enables robust motion transfer across diverse object categories and morphological variances without relying on category-specific templates.

## 3 METHODS

Our goal is to transfer articulated motion from a monocular video to arbitrary 3D characters. We take as input a static 3D target mesh $\mathcal{M}^{tgt}$ and a source monocular RGB video with paired foreground masks $\{I_t, M_t\}_{t=0}^T$, where $I_t$ is a frame from time $t$, and $M_t$ is obtained via off-the-shelf segmentation model (Kirillov et al., 2023). We aim to produce a temporally coherent sequence of deformed meshes $\{\mathcal{M}_t^{tgt}\}_{t=0}^T$ that faithfully reproduces the source motion.

We first encapsulate the target mesh with an Articulated-GS (Yao et al., 2025) representation with pose parameters (Sec. 3.1). We then parameterize morphology using learnable bone lengths, a global scale, and local Gaussian offsets (Sec. 3.2). This representation disentangles shape variation from pose dynamics. Finally, all shape and pose parameters are optimized jointly via differentiable rendering and dense semantic correspondences (Sec. 3.3–3.4), yielding semantically coherent motion aligned to the source. Fig. 2 illustrates the full pipeline.

### 3.1 ARTICULATED 3D GAUSSIAN SPLATTING FOR IMAGE-SPACE OPTIMIZATION

Retargeting motion from a monocular video typically requires estimating the 3D geometry of the source subject. However, inferring accurate 3D pose and shape from 2D inputs is inherently ambiguous. Reliance on these estimated 3D priors often introduces errors that propagate to the final result. We propose a direct optimization strategy to address this issue. We optimize the target character to align directly with the 2D source video observations. This approach bypasses the need for an explicit intermediate 3D representation of the source.

To this end, we employ Articulated 3D Gaussian Splatting (Articulated-GS) (Yao et al., 2025). This framework defines the target character using a single, unified canonical shape. We deform this time-invariant geometry via Linear Blend Skinning (LBS) to match the pose in each video frame. Critically, our optimization updates this single canonical shape to satisfy projection constraints across all time steps and camera views. This enforces geometric consistency throughout the entire motion sequence.

**Target Representation.** We represent the target character using a set of 3D Gaussians attached to a kinematic skeleton $\mathcal{T} = (\mathcal{J}, \mathcal{A})$, where $\mathcal{J}$ denotes the set of joints and $\mathcal{A} = \{A_j\}_{j \in \mathcal{J} \setminus \{j_r\}}$

Figure 2: **Overview of the morphology-adaptive articulated Gaussian splatting pipeline.** Given a target mesh, we parameterize it with deformable 3D Gaussians. A time-conditioned MLP ($f_{\text{MLP}}$) predicts skeletal transformations driven by input time embeddings. Crucially, our pipeline employs morphology adaptation (Sec. 3.2) to align the target's canonical structure, followed by LBS-based deformation (Sec. 3.1) for articulation. The framework is optimized end-to-end using differentiable rendering ($\mathcal{L}_{\text{render}}$) and semantic keypoint constraints ($\mathcal{L}_{\text{keypoint}}$) consistent with the source video.

maps each joint $j$ to its parent $A_j$, with $j_r$ being the root. Each Gaussian $G_i$ is parameterized by its mean $\boldsymbol{\mu}_i \in \mathbb{R}^3$, rotation $\boldsymbol{q}_i \in \mathbb{R}^4$, scale $\boldsymbol{s}_i \in \mathbb{R}^3$, opacity $\sigma_i \in [0, 1]$, and spherical harmonic coefficients $\mathcal{SH}_i \in \mathbb{R}^K$. Unlike previous works that initialize from sparse point clouds, we leverage the explicit geometry of the target mesh to initialize these Gaussian positions $\boldsymbol{\mu}_i$ (Sec. 3.2). For unrigged meshes, we employ automatic rigging methods (Xu et al., 2020; Zhang et al., 2025b) to establish the skeletal structure.

**Kinematic Deformation.** To capture temporal dynamics, a time-conditioned MLP, $f_{\text{MLP}}$, predicts the skeletal pose for each timestamp $t$. Given a sinusoidal time embedding $\text{emb}(t)$, the network outputs the root translation and relative joint rotations:

$$\left\{ \{\boldsymbol{\theta}_j^t\}_{j \in \mathcal{J}}, \ \boldsymbol{\delta}_{\text{global}}^t \right\} = f_{\text{MLP}}(\text{emb}(t)), \tag{1}$$

where $\boldsymbol{\theta}_j^t$ is the unit quaternion for joint $j$ and $\boldsymbol{\delta}_{\text{global}}^t$ is the global translation. These predictions drive the deformation of the canonical Gaussians. The deformed position $\boldsymbol{\mu}_i^t$ of Gaussian $i$ is computed via LBS:

$$\boldsymbol{\mu}_i^t = \boldsymbol{\delta}_{\text{global}}^t + \sum_{j \in \mathcal{J}} w_{ij} \mathbf{T}_j^t \bar{\boldsymbol{\mu}}_i, \quad \mathbf{T}_j^t = \prod_{k \in \text{P}(\text{root}, j)} \bar{\mathbf{T}}_k^t, \quad \bar{\mathbf{T}}_k^t = \begin{pmatrix} \mathbf{R}_k^t & \mathbf{J}_{A_k} - \mathbf{R}_k^t \mathbf{J}_{A_k} \\ 0 & 1 \end{pmatrix}. \tag{2}$$

Here, $\bar{\boldsymbol{\mu}}_i$ is the canonical center, $w_{ij}$ is the skinning weight, and $\mathbf{R}_k^t$ is the rotation matrix derived from $\boldsymbol{\theta}_k^t$. This formulation ensures that the Gaussians move coherently according to the skeletal hierarchy.

**Differentiable Rendering.** The deformed Gaussians are rasterized into 2D images to compute the optimization loss. For a viewpoint $\boldsymbol{v}$ and pixel $\boldsymbol{u}$, the color $\mathcal{C}(\boldsymbol{u})$ is derived via alpha compositing:

$$\mathcal{C}(\boldsymbol{u}) = \sum_{i \in \mathcal{N}} T_i \, \alpha_i \, \mathcal{SH}(\boldsymbol{sh}_i, \boldsymbol{v}), \quad T_i = \prod_{j=1}^{i-1} (1 - \alpha_j). \tag{3}$$

This differentiable rendering process allows us to backpropagate gradients from the 2D projection error directly to the 3D pose and shape parameters, bridging the domain gap between the 2D source and 3D target.

## 3.2 MORPHOLOGY-ADAPTIVE SHAPE PARAMETERIZATION

Standard Articulated-GS assumes a fixed skeletal topology, which restricts its ability to transfer motion between characters with differing limb proportions. To address this, we introduce a morphology-adaptive parameterization that explicitly disentangles structural variations from pose dynamics. In this paper, we use the term *morphology* to refer to the character's limb proportion, global body scale, and local shape details. By optimizing these time-invariant parameters alongside time-variant poses, our framework enables the target character to adapt its shape to the source motion while preserving kinematic coherence (Fig. 3 (b)).

**Learnable Bone Lengths.** We first relax the fixed skeleton constraint by assigning a learnable scalar length $\ell_b \in \mathbb{R}^+$ to each bone $b \in \mathcal{B}$. Given the unit direction vector $\boldsymbol{v}_b \in \mathbb{R}^3$ from a parent to a child

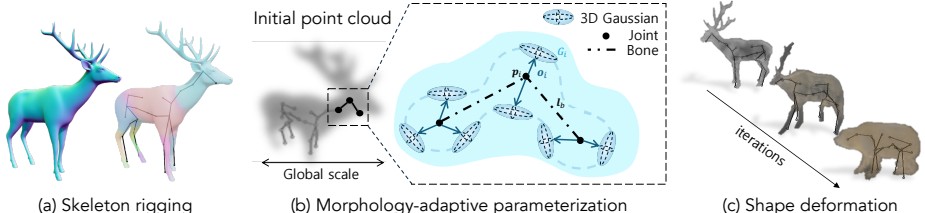

(a) Skeleton rigging    (b) Morphology-adaptive parameterization    (c) Shape deformation

Figure 3: **Deformable morphology parameterization.** **(a)** We initialize the target character with skeleton rigging, acquiring the topological structure and skinning weights. **(b)** Morphology-adaptive parameterization of structural variations. **(c)** During optimization, shape parameters deform the target's morphological structure to align with the morphology of the source.

joint, the rest-pose position of any joint $j$ is determined by the cumulative length of bones along the kinematic chain:

$$\mathbf{j}_{\text{rest}}(j) = \mathbf{j}_{\text{rest}}(j_{root}) + \sum_{b \in \text{P}(\text{root},j)} \ell_b \boldsymbol{v}_b. \tag{4}$$

This allows the skeleton to stretch or shrink segments (e.g., legs or arms) to match the source subject's proportions purely through optimization.

**Morphology-Aware Gaussian Initialization.** Crucially, the surface geometry must adapt to these skeletal changes. Instead of treating Gaussian positions as independent variables, we parameterize the mean $\boldsymbol{\mu}_i$ of each Gaussian $G_i$ relative to the underlying bone structure. We define $\boldsymbol{\mu}_i$ as a displacement from a skeleton-anchored reference point $\boldsymbol{p}_i$:

$$\boldsymbol{\mu}_i = \boldsymbol{p}_i + \boldsymbol{o}_i, \quad \text{where} \quad \boldsymbol{p}_i = \sum_{j \in \mathcal{J}} w_{ij} \, \mathbf{j}_{\text{rest}}(j), \tag{5}$$

where $\boldsymbol{p}_i$ represents the coarse geometry derived from joint positions $\mathbf{j}_{rest}(j)$ LBS weights $w_{ij}$, while the learnable offset $\boldsymbol{o}_i \in \mathbb{R}^3$ captures fine-grained local shape deviations. This formulation ensures that when bone lengths $\ell_b$ change, the associated Gaussians move coherently with the skeleton, preventing geometric artifacts.

**Global Scale and Canonical Shape.** Finally, to resolve the scale ambiguity inherent in monocular video, we introduce a global scaling factor $s_{global} \in \mathbb{R}^+$. This factor uniformly scales the entire morphology-parameterized character. The final canonical position $\bar{\boldsymbol{\mu}}_i$ used for deformation (Eq. 2) is obtained by:

$$\bar{\boldsymbol{\mu}}_i = s_{global} \cdot \boldsymbol{\mu}_i. \tag{6}$$

By jointly optimizing bone lengths ($\ell_b$), local offsets ($\boldsymbol{o}_i$), and global scale ($s_{global}$), our parameterization allows the target mesh to conform to the source's morphology while maintaining its original topological structure (Fig. 3 (c)).

**Discussion.** Our morphology parameterization provides a structural basis for mitigating the shape-pose ambiguity inherent in 2D-to-3D motion transfer. By explicitly decoupling global scale, skeletal lengths, and local offsets, our formulation promotes geometric identifiability under non-degenerate motion conditions, showing that morphological changes are distinguishable from pose dynamics. This disentanglement facilitates stable optimization by reducing the solution space to physically plausible configurations. We provide a detailed discussion on theoretical analysis in Appendix B.

### 3.3 TARGET-SOURCE DENSE SEMANTIC CORRESPONDENCE

While our proposed shape parameterization accounts for morphological differences, a key challenge in transferring articulated motion from 2D to 3D remains: *shape–pose ambiguity*. This refers to the inherent uncertainty in disentangling an object's underlying pose from its observation. Photometric loss provides essential low-level supervision, but relying on it alone may produce motion artifacts, as it captures only visual cues and lacks explicit semantic correspondences between characters. These artifacts can be mitigated by incorporating additional semantic cues, which help disambiguate overlapping projections, particularly when source and target morphologies differ.

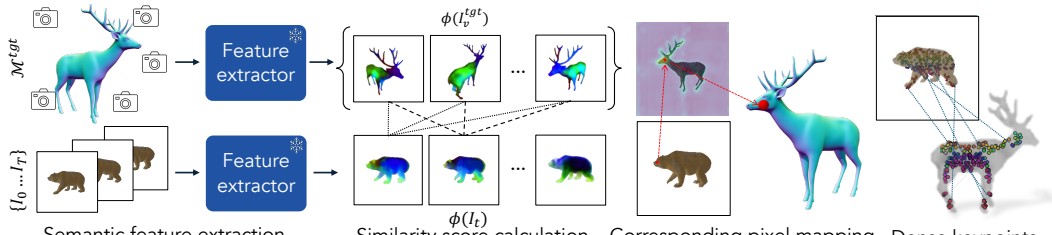

Figure 4: **Dense target-source correspondences matching.** We extract robust 2D-to-3D semantic correspondences by matching semantic features between source frames and rendered target views.

To address this, we establish robust 2D-3D semantic correspondences by leveraging pre-trained vision foundation models. Specifically, we utilize an orientation-sensitive feature extractor (Yang et al., 2020) that produces spatially consistent descriptors across varied poses and morphologies, then obtain dense pixel-to-vertex mappings through semantic feature matching between input images and target mesh renderings (Shtedritski et al., 2024). This provides automatic correspondence estimation without requiring manual registration or additional training.

The detailed pipeline of our dense correspondence extraction module is illustrated in Fig. 4. We first compute the similarity score of the dense semantic features extracted by the feature extractor $\phi(\cdot)$ from a source video frame $I_t$ with those from multiple rendered views $\{I_v^{\text{tgt}}\}$ of the target mesh $\mathcal{M}^{\text{tgt}}$. Then, given a source pixel $\boldsymbol{p} \in I_t$ with the extracted feature $\phi(I_t)$, we compute a pooled similarity score $\Sigma_{I_t}(\boldsymbol{p}, \boldsymbol{x}_k)$ for each vertex $\boldsymbol{x}_k \in \mathcal{M}^{\text{tgt}}$ as:

$$\Sigma_{I_t}(\boldsymbol{p}, \boldsymbol{x}_k) = \underset{v, \boldsymbol{x}_k \in \text{vis}(I_v^{\text{tgt}})}{\text{pool}} S\left(\phi(I_t)[\boldsymbol{p}], \phi(I_v^{\text{tgt}})[\pi_v(\boldsymbol{x}_k)]\right), \tag{7}$$

where $S(\cdot)$ denotes a cosine similarity, $\pi_v(\boldsymbol{x}_k)$ denotes the 2D projection of vertex $\boldsymbol{x}_k$ onto the rendered image $I_v^{\text{tgt}}$, and $\phi(I_v^{\text{tgt}})[\pi_v(\boldsymbol{x}_k)]$ is the corresponding feature vector at the 2D projected location. The operator $\text{pool}$ aggregates similarity scores via max-pooling across all $v$ target-rendered views where $\boldsymbol{x}_k$ is visible.

The best-matching 3D vertex $\tilde{\boldsymbol{x}}_{\boldsymbol{p},t}^{3D}$ for each pixel $\boldsymbol{p}$ in frame $t$ is obtained by selecting the vertex with the highest pooled similarity score:

$$\tilde{\boldsymbol{x}}_{\boldsymbol{p},t}^{3D} = \arg \max_{\boldsymbol{x}_k \in \mathcal{V}(\mathcal{M}^{\text{tgt}})} \Sigma_{I_t}(\boldsymbol{p}, \boldsymbol{x}_k), \tag{8}$$

where $\mathcal{V}(\mathcal{M}^{\text{tgt}})$ denotes the set of vertices of the target mesh. These retrieved 3D points $\tilde{\boldsymbol{x}}_{\boldsymbol{p},t}^{3D}$ serve as semantic keypoints, providing supervision to guide semantic structure alignment of cross-modality during optimization, as the keypoint loss $L_{\text{keypoint}}$ (Sec. 3.4).

## 3.4 OPTIMIZATION

As formalized in Eq. 1 and visualized in Fig. 2, our primary objective is to recover the target mesh's time-varying skeletal pose parameters aligned with the source motion, relying solely on 2D observations without ground-truth 3D annotations or any form of pose template prior. The entire framework, composed of morphology-parameterized articulated Gaussians, is optimized end-to-end by minimizing a composite loss function. Our optimization objective combines photometric reconstruction, semantic correspondence, and multiple regularization terms: $\mathcal{L}_{\text{total}} = \lambda_{\text{render}}\mathcal{L}_{\text{render}} + \lambda_{\text{keypoint}}\mathcal{L}_{\text{keypoint}} + \lambda_{\text{reg}}\mathcal{L}_{\text{reg}}$, where the weights balance their respective contributions.

The render loss enforces photometric consistency between the rendered frame $\hat{I}_t$ (from Eq. 3) and the source frame $I_t$ by combining an $\ell_1$ term with a SSIM (Wang et al., 2004) term:

$$\mathcal{L}_{\text{render}} = \sum_{t=0}^{T} \left[(1 - \lambda_{\text{dSSIM}}) \left\|\hat{I}_t - I_t\right\|_1 + \lambda_{\text{dSSIM}}\left(1 - \text{SSIM}(\hat{I}_t, I_t)\right)\right]. \tag{9}$$

The keypoint loss supervises geometric alignment by minimizing projection error between source image pixels and their matched 3D vertices derived from dense semantic correspondences:

$$\mathcal{L}_{\text{keypoint}} = \sum_{t=0}^{T} \sum_{\boldsymbol{p} \in \mathcal{P}_t} \left\|\boldsymbol{p} - \pi_t\left(\tilde{\boldsymbol{x}}_{\boldsymbol{p},t}^{3D}\right)\right\|_2, \tag{10}$$

Table 1: **Quantitative evaluation on Mixamo and DT4D datasets.** Our method consistently outperforms all baselines across diverse categories. Results are averaged across scenes, with per-scene results in the Appendix.

| | Mixamo | | DT4D-Quadrupeds | | DT4D-Others | |
|---|---|---|---|---|---|---|
| | PMD ↓ | FID ↓ | PMD ↓ | FID ↓ | PMD ↓ | FID ↓ |
| SPT$^+$ | 0.0029 | 0.0366 | - | - | - | - |
| NPR$^+$ | 0.0099 | 0.0551 | 0.0032 | 0.0669 | - | - |
| Transfer4D | 0.0084 | 0.0855 | 0.0058 | 0.0505 | 0.0133 | 0.0805 |
| Ours | **0.0028** | **0.0304** | **0.0018** | **0.0171** | **0.0023** | **0.0124** |

where $\tilde{x}_{p,t}^{3D}$ is the best-matching 3D vertex obtained via Eq. 8, and $\mathcal{P}_t$ represents sampled foreground pixels. Finally, $\mathcal{L}_{\text{reg}}$ comprises multiple regularization terms that encourage temporal smoothness and geometric consistency (detailed formulations provided in the Appendix).

# 4 EXPERIMENTS

## 4.1 DATASETS AND IMPLEMENTATIONS

**Datasets.** We evaluate our approach on mesh-animation pairs sampled from DeformingThings-4D (DT4D) (Li et al., 2021) and Mixamo (Adobe). From DT4D, we select 20 animation pairs spanning diverse animal categories of quadrupeds and non-quadrupeds exhibiting varied motions. From Mixamo, we utilize 12 humanoid mesh-animation pairs across different character models and motion types. To simulate a *casually captured* monocular video scenario, we render each source animation using a single camera with constrained movement (±30° angular range), generating input frames with corresponding ground-truth 3D target mesh animations. We further conduct qualitative evaluation on real-world videos sourced from the DAVIS dataset (Perazzi et al., 2016) and two publicly available online videos (Daley, n.d.; Nicky Pe, n.d.), as well as 2D-to-2D motion transfer scenarios using additional synthetic sequences (Pumarola et al., 2021; Liu et al., 2024). Details on dataset preparation and configuration are provided in the Appendix.

**Implementation details.** We employ a two-stage optimization strategy that first performs global alignment of scale and translation, then jointly refines local pose and shape parameters (bone length, Gaussians) to adapt morphology while preserving essential motion characteristics. All experiments use the Adam optimizer (Kingma & Ba, 2014) with adaptive learning rates over 10k iterations. Our method achieves efficient optimization, completing training in under 10 minutes on a single RTX 4090 GPU. Detailed hyperparameter specifications are provided in the Appendix.

## 4.2 2D-TO-3D MOTION TRANSFER

**Baselines and metrics.** We compare our method against two baseline categories: *composite pipelines* combining 2D-to-3D reconstruction with 3D motion transfer, and a template-free optimization-based approach (Transfer4D (Maheshwari et al., 2023)). For composite baselines, we adopt a two-stage setup with mesh reconstruction followed by motion transfer using SPT (Liao et al., 2022) and NPR (Yoo et al., 2024), denoted as SPT$^+$ and NPR$^+$ (see Appendix for baseline implementation details). SPT$^+$ is evaluated only on humanoid motion, as the original method was designed and tested on stylized human characters. Transfer4D performs motion retargeting by extracting skeletal structure from RGB-D input. On datasets with non-quadruped animals, where parametric templates of reconstruction methods are not applicable, we compare only to Transfer4D.

We quantify motion transfer by comparing the retargeted and ground-truth mesh sequences. Consistent with prior work (Liao et al., 2022; Yoo et al., 2024), we adopt Point-wise Mesh Distance (PMD) to measure per-vertex accuracy and Fréchet Inception Distance (FID) (Heusel et al., 2017) to assess perceptual fidelity. To compute FID, both ground-truth and retargeted animations are rendered from 12 viewpoints and their image distributions are compared.

**Comparison results.** We evaluate our method and baselines on DT4D and Mixamo datasets. As shown in Tab. 1, our approach achieves superior performance on both PMD and FID metrics. These results show that our approach achieves strong performance in a data-efficient manner, relying only on direct optimization without explicit 3D supervision. On non-quadrupeds (DT4D-Others), we

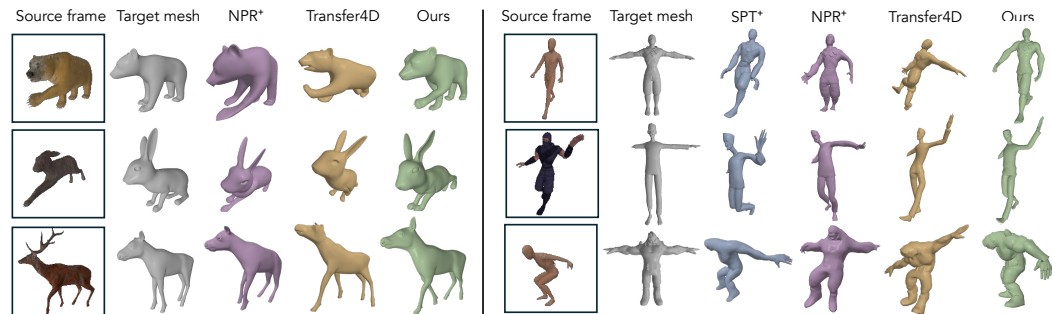

Figure 5: **Qualitative results on Mixamo and DT4D-Quadruped datasets.** Our method shows superior pose alignment compared to baselines across diverse objects. Refer to the supplementary video for full animation.

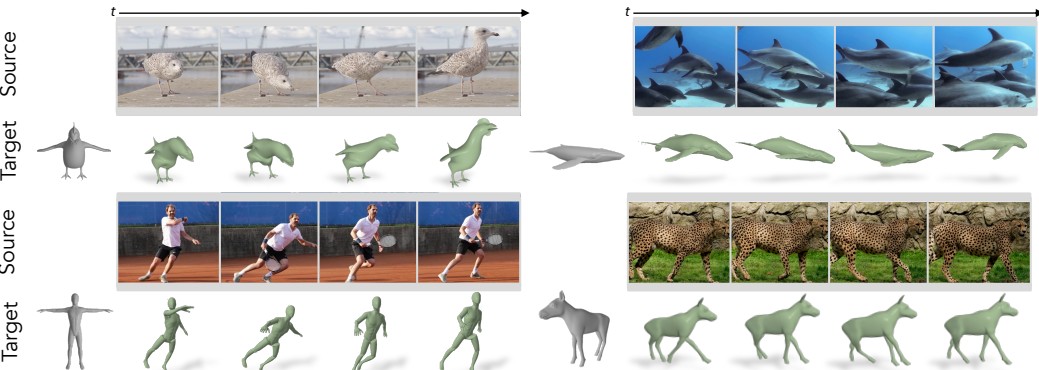

Figure 6: **Qualitative results on real-world datasets.** Our motion transfer pipeline effectively transfers motion from both synthetic and real-world videos in a category-agnostic manner.

significantly outperform Transfer4D even without depth input, demonstrating strong performance beyond parametric model categories.

Fig. 5 demonstrates that our method preserves the target shape and transfers motion faithfully, while baselines often produce distorted shapes by estimating incorrect transformation (Liao et al., 2022; Maheshwari et al., 2023) or relying on predicted surface Jacobians (Yoo et al., 2024). This shape fidelity is attributed to our morphology-parameterization, which we also analyze in Sec. 4.3.

**Qualitative results on real-world videos.** To evaluate real-world applicability, we apply our method to in-the-wild monocular videos featuring diverse animal categories with complex backgrounds and occlusions. These noisy or open-domain scenarios represent cases where obtaining corresponding 3D animations is challenging. As shown in Fig. 6, our approach successfully transfers motion across these varied scenarios while preserving target mesh structure and proportions. These results demonstrate effective motion transfer directly from monocular input without requiring 3D motion generation, highlighting the practical value of our 2D-grounded motion transfer approach.

### 4.3 ABLATION STUDY

We ablate key components of our framework in Tab. 2 and Fig. 7. Removing the rendering loss severely degrades performance (PMD $\uparrow \sim 5\times$), indicating it as the primary driver of motion transfer, while the keypoint loss adds complementary semantic guidance. Fig. 7 shows that dropping the keypoint loss yields suboptimal transfers due to unresolved shape-pose ambiguities.

Excluding our shape parameterization (bone lengths $l_b$, Gaussian means $\mu$, global scale $s_{global}$) causes distorted geometry and misaligned orientations, especially under large morphological differences. With shape parameters fixed, global translation lowers render loss by pulling the object toward the camera, partially recovering motion but distorting orientation and pose (Fig. 7; see supplementary videos). Overall, adding each component yields consistent gains (Tab. 2), confirming their complementary roles to enhance robustness. Extended ablation studies appear in the Appendix.

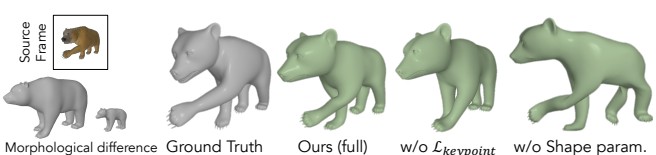

Figure 7: **Qualitative ablations.** Keypoint loss complements motion details and accuracy. Excluding shape parameters induces severe geometric artifacts for large morphological variation.

Table 2: **Quantitative evaluation of component contributions.**

| Ablation | PMD ($\downarrow$) | FID ($\downarrow$) |
|---|---|---|
| Full Model | **0.0018** | **0.0171** |
| *w/o* $\mathcal{L}_{render}$ | 0.0090 | 0.0463 |
| *w/o* Shape param. | 0.0047 | 0.0747 |
| *w/o* $\mu$ update | 0.0039 | 0.0552 |
| *w/o* $l_b$ & $s_{global}$ update | 0.0040 | 0.0488 |
| *w/o* $\mathcal{L}_{keypoint}$ | 0.0031 | 0.0252 |

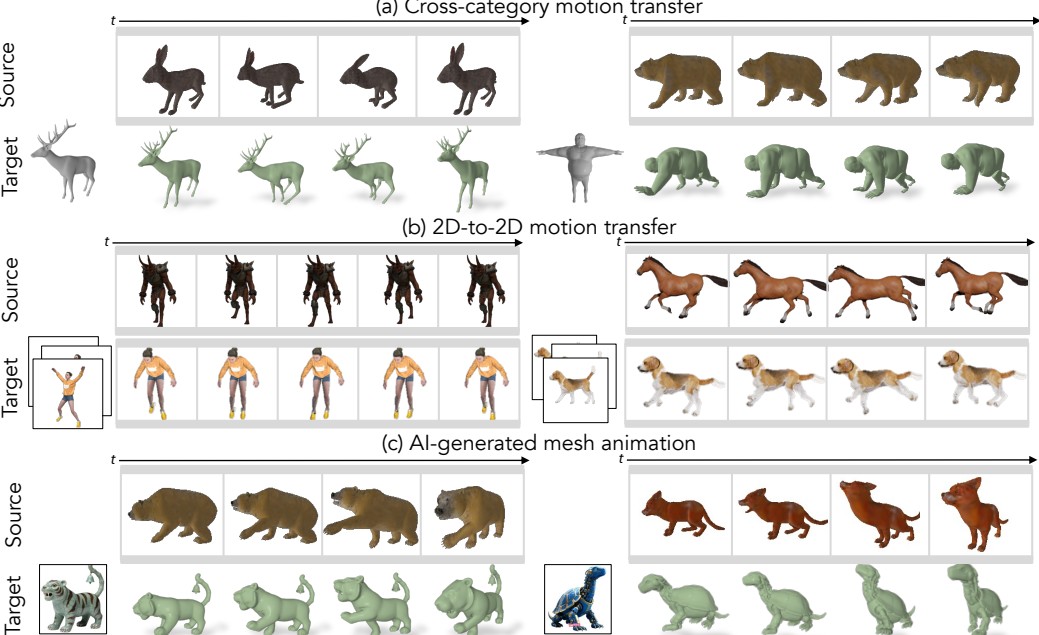

Figure 8: **Results on diverse applications.** Our method transfers motion for (a) cross-category source-target pairs, (b) 2D-to-2D videos, and (c) AI-generated mesh animations.

## 4.4 DIVERSE APPLICATION SCENARIOS

**Cross-category motion transfer.** Our method demonstrates strong generalization across diverse categories, as shown in Fig. 8 (a). We successfully transfer motion between different animal species (rabbit-to-deer) and even across broader categories (animal-to-human). This flexibility stems from our universal optimization approach that does not rely on category-specific skeletal structures or explicit category matching between source and target.

**2D-to-2D motion transfer.** A key advantage of our method is its representation-agnostic applicability across articulated 3D assets. While primarily demonstrated on mesh targets, our framework seamlessly extends to Gaussian-based 3D representation without modification of core design. Fig. 8(b) shows motion transfer to 3DGS reconstructed from multi-view images (Yao et al., 2025), enabling video-to-video transfer when both source and target originate from RGB sequences. Together, these results yield a single, category-agnostic framework that operates consistently across varied 3D representations.

**AI-generated mesh animation.** Another interesting application is animating meshes synthesized by generative models. As shown in Fig. 8 (c), we achieve effective motion transfer using meshes generated from an off-the-shelf image-to-mesh model (Zhao et al., 2025). This demonstrates the versatility of our approach to meshes from diverse sources, supporting modern content creation workflows that increasingly incorporate AI-generated assets.

## 5 DISCUSSION

We introduce **CAMO**, a framework that transfers motion from monocular videos to 3D assets without relying on category-specific templates. By reformulating motion retargeting as an efficient morphology-adaptive optimization on articulated Gaussian splats, our method avoids error accumulation in traditional reconstruct-then-retarget pipelines without any 3D supervision or large datasets. The integration of morphology-adaptive modeling and semantic correspondences provides complementary cues that reduce shape-pose ambiguities and enable broad applicability across different skeletal structures and 3D representations.

**Limitations and future work.** While CAMO achieves robust category-agnostic motion transfer, the current morphology-adaptive formulation is limited to articulated kinematic structures. This restricts its ability to capture richer non-rigid dynamics such as soft-tissue deformation or secondary motion (e.g., hair dynamics, tail sway). Beyond these kinematic limitations, our framework currently prioritizes visual motion transfer rather than enforcing full physical plausibility. A promising direction is to augment our optimization with physically grounded constraints, such as Jacobian-space motion consistency and contact-aware regularization. Another promising avenue for future work is to enrich the framework with additional geometric cues, such as monocular depth predictors or generative 3D priors. These sources of structure-aware regularization could improve robustness in complex scenes or under limited camera motion, further extending the applicability of our approach.

## Ethics Statement

This work presents a novel framework for motion synthesis, intended for beneficial applications in digital content creation, robotics, and virtual reality. While our research focuses on advancing 3D motion synthesis techniques, we acknowledge the potential risks associated with generative technologies, such as the creation of deceptive content or deepfakes. To address privacy and data ethics, all experiments were conducted using publicly available benchmark datasets (e.g., Mixamo (Adobe), DT4D (Li et al., 2021), DAVIS (Perazzi et al., 2016)) and open-license video resources (Pexels). All data sources were utilized in strict accordance with their respective licenses and usage guidelines. Technically, our method is designed to transfer motion rather than identity; however, we emphasize that responsible deployment is essential to mitigate misuse, and we advocate for adherence to ethical guidelines and legal frameworks.

## Reproducibility Statement

We are committed to ensuring the reproducibility of our results. We provide detailed descriptions of our pipeline, including articulated 3D Gaussian Splatting, morphology-adaptive parameterization, and dense semantic keypoint correspondence in Sec. 3. Furthermore, detailed objective functions, data preprocessing steps, specific hyperparameter settings, and additional ablation studies verifying the robustness of our method are comprehensively reported in Appendix A.1 and A.2.

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

# A  TECHNICAL APPENDICES AND SUPPLEMENTARY MATERIAL

This appendix provides additional implementation details, ablations, and extended results supporting the main paper. The overall structure for the Appendices is as follows:

- **Datasets and baselines** (Sec. A.1)
- **Implementational details** (Sec. A.2)
- **Ablation on design choices** (Sec. A.3)
    - Shape parameterization
    - Dense keypoint loss
    - Rigging modules
    - Geometry-aware semantic features
- **Performance analysis** (Sec. A.4)
    - Performance on diverse morphological variations
    - Performance on different motion scales
    - Performance on challenging cases
    - Failure case analysis
    - Computational analysis (New)
    - Analysis on camera initialization (New)
    - Robustness under occlusion (New)
    - Multi-view scalability (New)
    - Robustness against fast motion and motion blur (New)
- **Extended Tables and Qualitative Videos** (Sec. A.5)
- **Comparison with generative 4D pipelines.** (New) (Sec. A.6)

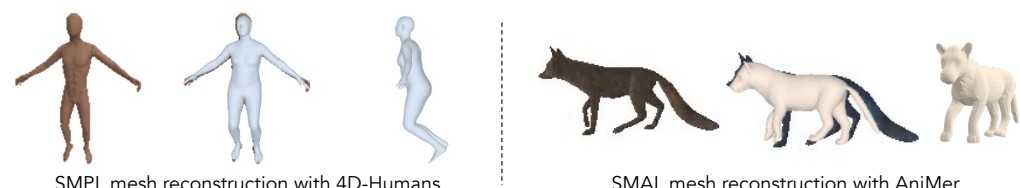

SMPL mesh reconstruction with 4D-Humans          SMAL mesh reconstruction with AniMer

Figure 9: **Intermediate mesh reconstruction with template-based 3D pose and shape estimation models.**

## A.1 DATASETS AND BASELINES

**Synthetic datasets.** DeformingThings4D (DT4D) (Li et al., 2021) is a large-scale synthetic dataset of non-rigidly deforming objects, featuring 1,972 animation sequences across 147 characters from 31 categories made by CG experts. We specifically select animal motion sequences (DT4D-animals) for evaluation. We collect 20 pairs of animations, each pair sharing identical pose parameters but differing in character shape. For humanoid characters, we utilize Mixamo (Adobe) to acquire 12 character-motion pairs of discrete motions. Example datasets URL for animal and humanoid datasets are provided in our `index.html`.

To generate the monocular video, we render source animations from the DT4D and Mixamo datasets at a resolution of 256×256 using PyTorch3D's `PerspectiveCamera`, ensuring consistent viewpoint changes by varying camera azimuth within ±30°. Untextured DT4D sequences are textured with texture maps generated from TexPainter (Zhang et al., 2024b) to improve visual realism.

**Real-world videos.** For real-world videos collected from different sources (Daley, n.d.; Nicky Pe, n.d.; Perazzi et al., 2016), we clip and resize the videos at a resolution of 256x256. While synthetic datasets provide ground-truth camera configurations and global orientation alignment between source and target sequences, real-world videos lack such information; thus, we assume a fixed camera for real-world videos. To align the 3D target mesh with the source video's object orientation and scale for motion transfer, we adopt a render-and-compare strategy guided by semantic correspondences. Specifically, we pre-render target mesh with candidate camera poses and evaluate each pose by calculating patch-wise feature cosine similarity to the source frame. The camera pose yielding the maximum similarity serves as our initial alignment, providing a stable and semantically grounded initialization for subsequent optimization.

**Implementation of Composite baselines.** As described in Sec. 4 in main paper, we compare our method with *composite pipelines* that first reconstruct 3D source meshes from 2D videos, followed by 3D-to-3D motion retargeting. Intermediate reconstructions are obtained by fitting parametric templates to each video frame: SMPL (Loper et al., 2015) for humans and SMAL (Zuffi et al., 2017) for quadrupeds (see Fig.9 for reconstruction examples).

For humanoid motion transfer on the Mixamo dataset, we first extract SMPL meshes using 4D-Humans (Goel et al., 2023), then apply SPT (Liao et al., 2022) and NPR (Yoo et al., 2024) with pretrained checkpoints based on the SMPL model. For quadruped experiments on DT4D, we train NPR's pose extractor and shape applier modules using SMAL meshes reconstructed from monocular videos via AniMer (Lyu et al., 2024).

## A.2 IMPLEMENTATIONAL DETAILS

**Skeletal motion field.** The skeletal motion field is parameterized by MLPs. Temporal inputs are first processed using sinusoidal embeddings (13-dimensional), and subsequently passed through a two-layer embedding network producing a 30-dimensional temporal representation. This representation is then fed into an 8-layer MLP featuring 256 hidden units and a skip connection at the fourth layer. The MLP outputs are then directed to a 2-layer global translation head predicting 3D translation vectors, and a 2-layer joint rotation head predicting normalized quaternions. Together, these outputs define SE(3) transformation governing the skeletal motion.

**Motion regularizers.** Our method employs four distinct motion regularizers to ensure stable and plausible motion. To prevent excessive motions during early training, we apply an $L_1$ penalty jointly

to global translations and joint rotations:

$$\mathcal{L}_{\text{trans}} = \lambda_{\text{trans}} \frac{\|\boldsymbol{\delta}_{\text{global}}^t\|_1 + \sum_{j=1}^{J} \|r_j^t\|_1}{J}, \tag{11}$$

where $\lambda_{\text{trans}}$ is the regularization weight, $\boldsymbol{\delta}_{\text{global}}^t$ the global translation vector at frame $t$, $r_j^t$ the rotation angle for joint $j$ at frame t, and $J$ the number of joints.

To enforce temporal smoothness, we additionally penalize frame-to-frame motion:

$$\mathcal{L}_{\text{smooth}} = \lambda_{\text{smooth}} \left( \sum_{j=1}^{J} \|r_j^t - r_j^{t-1}\|_1 + \|\boldsymbol{\delta}_{\text{global}}^t - \boldsymbol{\delta}_{\text{global}}^{t-1}\|_1 \right), \tag{12}$$

where $\lambda_{\text{smooth}}$ is the smoothness weight.

Following Yao et al. (Yao et al., 2025), we impose 2D projection constraints on 3D points sampled along the articulated skeleton. First, we extract 2D skeleton points $p_{\text{skeleton}}^t$ from the source foreground mask $M_{\text{src}}^t$ via a morphological thinning algorithm (Zhang & Suen, 1984). Then, at each frame $t$, we sample a set of 3D points $c^t$ on the deformed skeleton, project them into the image plane using the camera projection $\pi_t$, and penalize their misalignment to the 2D skeleton:

$$\mathcal{L}_{\text{chamf}} = \lambda_{\text{chamf}} \, \text{CD}_{\ell_1} \left( p_{\text{skeleton}}^t, \, \pi_t(c^t) \right), \tag{13}$$

where $\text{CD}_{\ell_1}$ denotes the Chamfer distance (Fan et al., 2017) under the $\ell_1$ norm, and $\lambda_{\text{chamf}}$ is a hyperparameter that controls the regularization strength.

To ensure each joint remains within its assigned skinning region, we penalize the mean squared error between the deformed joint positions **j** and the centroids of their corresponding Gaussian groups, computed as weighted averages of the Gaussian means $\mu$ with normalized skinning weights:

$$\mathcal{L}_{\text{skin}} = \lambda_{\text{skin}} \sum_{j=1}^{J} \|\sum_{i=1}^{N} \tilde{w}_{ij}^{\top} \mu_i \, - \, \mathcal{J}_j\|^2, \quad \tilde{w}_{ij} = \frac{w_{ij}}{\sum_{i'=1}^{N} w_{i'j}}, \tag{14}$$

where $w \in \mathbb{R}^{N \times J}$ are the LBS skinning weights, $\mu \in \mathbb{R}^{N \times 3}$ the Gaussian mean positions, $\mathcal{J} \in \mathbb{R}^{J \times 3}$ the joint positions, and $\lambda_{\text{skin}}$ the skinning regularization weight.

**Training details.** As described in Sec. 3.4 of the main paper, we balance the rendering and keypoint losses with $\lambda_{\text{render}} = 1.0$ and $\lambda_{\text{keypoint}} = 0.001$. The motion regularization weights are set to $\lambda_{\text{transform}} = 0.005$, $\lambda_{\text{smooth}} = 0.001$, $\lambda_{\text{chamf}} = 0.0001$, and $\lambda_{\text{skin}} = 0.1$.

Training follows a two-stage schedule over 10K iterations using Adam optimizer (Kingma & Ba, 2014). The first 500 iterations optimize only global scale, bone length, and global translation for stable initialization. Subsequently, all parameters including shape parameters are jointly optimized with exponential learning rate decay.

We employ differentiated update frequencies based on parameter characteristics. Frame-specific parameters including articulated 3D Gaussians and local motion heads are updated per frame to capture temporal details. Shape parameters such as bone length and global scale, and the global translation are updated every 10 frames to maintain cross-frame consistency and motion smoothness.

### A.3 ABLATION ON DESIGN CHOICES

#### A.3.1 SHAPE PARAMETERIZATION

Shape parameters are essential for accurately capturing both global and local motion dynamics and ensuring consistent spatial orientation (Sec. 4.3). Inadequate scale regularization causes temporal drift toward the camera, where optimization compensates for scale discrepancies through global translation (shown in the supplementary videos). This compensation disrupts orientation estimation and motion coherence. In contrast, our complete formulation with comprehensive shape parameters preserves geometric consistency and produces stable motion reconstructions.

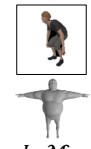 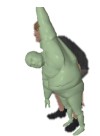 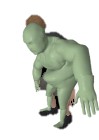

$I_t, \mathcal{M}_{tgt}$    w/o $L_{keyp}$    Ours (full)

Figure 10: **Qualitative ablation on keypoint loss.** Keypoint loss $L_{\text{keyp}}$ helps ensure correct motion reconstruction by resolving ambiguity where limbs overlap in the source video frame.

Table 3: **Ablation on keypoint confidence thresholding.** We observe that PMD and FID decrease slightly with higher thresholds, reflecting improved performance, yet remain stable across the full range.

| Thr. | PMD ↓ | FID ↓ |
|---|---|---|
| 0.0 | 0.0020 | 0.0178 |
| 0.7 | 0.0019 | 0.0179 |
| 0.9 | 0.0018 | 0.0171 |

Table 4: **Ablation on number of keypoints.** While 1K points yield slight improvements, performance remains comparable even at sparse points (#50).

| #Keyp. | PMD ↓ | FID ↓ |
|---|---|---|
| 50 | 0.0020 | 0.0191 |
| 100 | 0.0020 | 0.0185 |
| 500 | 0.0019 | 0.0179 |
| 1000 | 0.0018 | 0.0171 |
| 1500 | 0.0020 | 0.0185 |

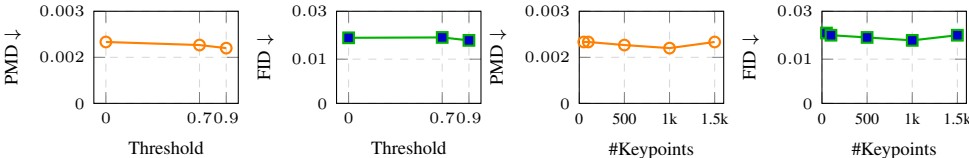

Figure 11: **Ablations on semantic keypoints.** Left two images: effects of confidence thresholds. Right two images: effects of keypoint counts. The results demonstrate robust performance across these configurations.

### A.3.2 DENSE KEYPOINT LOSS

As described in Sec.4.3, eliminating dense correspondence guidance leads to misaligned motion cues and misperception of semantic parts. Results illustrated in Fig.10 demonstrate that our dense semantic correspondence effectively encodes object-level semantics, enabling spatially consistent and semantically faithful motion generation.

We sample 1K keypoints with confidence above 90% to minimize the effect of outliers. Tab. 3 and Fig. 11 show that performance remains consistent across different confidence thresholds, demonstrating robustness to noisy correspondences. Tab. 4 and Fig. 11 suggest stable behavior of keypoint density effects across guidance densities, with slight improvement on 1K points.

### A.3.3 RIGGING MODULE ABLATION

We evaluate the impact of rigging quality on our DT4D dataset, including quadrupeds and non-quadrupeds, by comparing three rigging modules: RigNet (Xu et al., 2020), MagicArticulate (Song et al., 2025b), and UniRig (Zhang et al., 2025b). Enhanced rigging priors generally improve performance, as shown in Tab. 5 (DT4D-sub). We observe that skinning weight quality significantly affects results. While MagicArticulate and UniRig perform well on the subset, their performance varies on the full dataset, particularly for large motions (Tab. 5, DT4D-all). These results demonstrate the importance of high-quality skinning weights and suggest potential benefits from incorporating adaptive skinning refinement mechanisms.

### A.3.4 GEOMETRY-AWARE SEMANTIC FEATURES

Distinguishing geometrical differences (e.g., left/right limbs) is crucial for accurate motion transfer. We utilize a pretrained geometry-aware semantic feature extraction module (Yang et al., 2020) for dense correspondence matching. Tab. 6 ablates this design choice, comparing motion transfer performance when using alternative pretrained semantic features from foundation models (Stable Diffusion (Rombach et al., 2022) and DINOv2 (Oquab et al., 2023)) for correspondence matching. Performance significantly drops with SD or DINOv2 features, confirming the effectiveness of geometry-aware features for motion transfer tasks.

Table 5: **Ablation on rigging methods.** We compare performance on the DT4D dataset, where DT4D-sub consists of scenes with relatively small motions. 'CAMO+Best' denotes results using the optimal method for each scene.

|  | DT4D-sub | | DT4D-all | |
| --- | --- | --- | --- | --- |
| Method | PMD ↓ | FID ↓ | PMD ↓ | FID ↓ |
| CAMO + RigNet | 0.0018 | 0.0153 | 0.0019 | 0.0159 |
| CAMO + MagicArticulate | 0.0016 | 0.0094 | 0.0021 | 0.0117 |
| CAMO + UniRig | 0.0015 | 0.0110 | 0.0026 | 0.0168 |
| CAMO + Best | 0.0012 | 0.0086 | 0.0015 | 0.0108 |

Table 6: **Ablation on semantic features.** Our framework achieves best performance with geometry-aware semantic features by distinguishing relationships between body parts. This provides superior structural guidance compared to standard 2D feature extractors.

| Method | PMD ↓ | FID ↓ |
| --- | --- | --- |
| CAMO + Stable Diffusion | 0.0028 | 0.0561 |
| CAMO + DINOv2 | 0.0025 | 0.0191 |
| CAMO + Geo-Aware | 0.0018 | 0.0171 |

Table 7: **Performance on DT4D under morphological variations.** Shape differences increase substantially across groups (up to 341% from Low to High), yet performance degrades gracefully, highlighting robustness to diverse geometric discrepancies.

|  | Shape Differences | PMD ↓ | FID ↓ |
| --- | --- | --- | --- |
| Low | $0.00084 \pm 0.00036$ | 0.0012 | 0.0091 |
| Med | $0.00211 \pm 0.00016$ | 0.0017 | 0.0205 |
| High | $0.00287 \pm 0.00036$ | 0.0026 | 0.0218 |

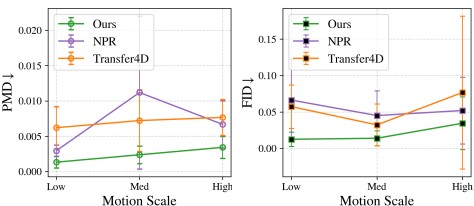

Figure 12: **Performance on DT4D by motion scale.** Our method outperforms baselines across all levels, with minimal degradation on large motions.

## A.4 PERFORMANCE ANALYSIS

### A.4.1 PERFORMANCE ACROSS MORPHOLOGICAL VARIATIONS

Our dataset encompasses diverse morphological differences between source and target subjects. We quantify these variations using two metrics: (1) *global scale* measured by mesh volume ratio to capture overall size differences, and (2) *shape distance* measured by Chamfer Distance (CD) on normalized meshes to assess geometric variations independent of scale. The dataset spans volume ratios from 1.08× to 16.00× and shape distances from 0.0004 to 0.0023, enabling comprehensive evaluation across morphological diversity.

Regarding global size, we find no correlation between performance and global scale differences. Dividing our dataset into three groups by scale magnitude, low and high groups achieve similar mean PMD (0.00150 vs 0.00136). Note that this requires the target mesh to initially lie within the camera frustum for valid optimization signals. Regarding shape differences, we categorize source-target pairs into three groups by shape distance. As shown in Tab. 7, our method achieves optimal results with minimal morphological differences while maintaining robust performance under considerable shape variations.

### A.4.2 PERFORMANCE ON DIFFERENT MOTION SCALES

We define motion magnitude as the maximum average vertex displacement from the first frame across the sequence, computed in normalized coordinate space for cross-mesh comparability. Our dataset spans diverse motion scales (min: 0.03, max: 0.23, avg: 0.11), which we categorized into three distinct groups ranging from small to large motion. Fig. 12 demonstrates consistent performance across motion scales, confirming robustness to motion scale variations. This robustness stems from our time-conditioned MLP jointly optimizing across frames to capture global trajectories and temporal dependencies, while the joint-rotation head provides frame-specific refinements, maintaining global coherence with localized flexibility.

### A.4.3 PERFORMANCE ON CHALLENGING CASES

We evaluate our method on two challenging scenarios that can potentially compromise performance: thin geometric structures and long video sequences.

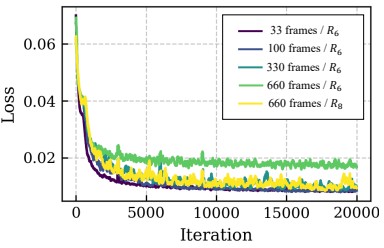

Figure 13: **Qualitative results on challenging case with thin structure.** Our method achieves robust performance on characters with thin structures, which may pose fundamental difficulties in motion transfer.

Table 8: **Quantitative evaluation of temporal capacity.** Values in parentheses indicate frequency bands ($L$). We observe that increasing $L$ effectively handles extended sequences, mitigating performance degradation.

| Frames (Temporal PE Bands) | PMD $\downarrow$ | FID $\downarrow$ |
|---|---|---|
| 100 ($R_6$) | **0.0010** | **0.0034** |
| 330 ($R_6$) | 0.0008 | 0.0060 |
| 660 ($R_6$) | 0.0012 | 0.0125 |
| 660 ($R_8$) | **0.0010** | 0.0047 |

Figure 14: **Loss curves vs. number of frames.** $R_L$ denotes temporal resolution with $L$ frequency bands for positional encoding.

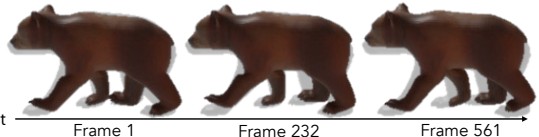

Figure 15: **Visualization of temporal drift.** We compare our result with the Ground Truth (shown as a gray silhouette underneath). While the method maintains high stability across the 660-frame sequence, slight drift accumulates in later frames (e.g., Frame 561), revealing the underlying GT.

First, challenging structures, such as thin bird wings, present difficulties for both visual guidance and mesh deformation. Their 2D projected regions cover only a few pixels, yielding limited visual cues, while their slender geometry is easily distorted during deformation. As shown in Fig.13, our method robustly addresses these geometrically challenging scenarios within reasonable performance. This robustness is enabled by 2D skeletal projection constraints and temporal smoothness regularization, which jointly enforce motion consistency across frames (Sec.A.2).

To support substantially longer sequences, our model can be scaled in two ways: (i) increasing the MLP capacity, or (ii) partitioning the video into temporal segments and optimizing a dedicated motion-field MLP per segment. This segmentation strategy effectively prevents error accumulation and maintains stable optimization over long durations. Our analysis of the model's behavior across varying sequence lengths validates the rationale behind this approach. While performance remains stable at 300 frames (PMD: 0.0010), it degrades for sequences exceeding 600 frames (PMD: 0.0012) as the mapping of time embeddings to complex motions demands greater representational capability. Increasing the frequency bands of sinusoidal positional encoding (e.g., $L = 6 \rightarrow 8$) restores optimization quality (Tab. 8, Fig. 14). However, naive extension eventually leads to temporal drift (Fig. 15), as the finite capacity of a fixed-size MLP saturates against the complexity of extremely long trajectories.

### A.4.4 FAILURE CASE ANALYSIS

Despite the robust performance, CAMO exhibits limitations when faced with significant occlusion or ambiguous left-right limb distinction in the source video, leading to less faithful motion transfer. Specifically, Fig. 16 (a) illustrates a failure case where motion quality degrades due to an unclear differentiation of the left and right legs. Fig. 16 (b) and Fig. 17 highlight performance degradation attributed to extensive self-occlusion, where insufficient visual information hinders accurate motion reconstruction.

### A.4.5 COMPUTATIONAL ANALYSIS

Figure 18 demonstrates the stability and efficiency of our framework. As illustrated in Fig. 18a, the optimization duration remains remarkably stable across varying input complexities. Despite variations in vertex count (avg. 12K, max. 19K) and sequence length (avg. 46, max. 140 frames), the process consistently completes within 4–5 minutes (median $<$ 4.6 minutes). Fig. 18b further shows

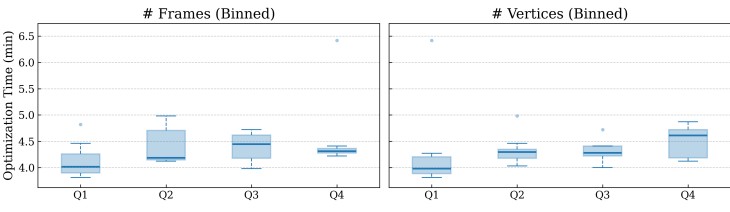

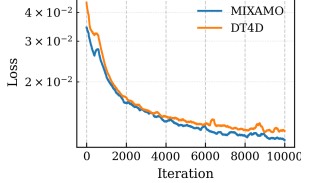

Figure 16: **Failure cases.** Representative failure cases include misperception of geometric semantics leading to left-right confusion (a) and pose estimation errors due to severe occlusion in the source video (b).

Figure 17: **Analysis of self-occlusion.** We investigate the impact of severe self-occlusion across varying camera angles. As demonstrated in the $0°$ camera angle, the method fails when the moving limb is completely occluded from the camera's perspective.

(a) Optimization time in accordance with frames and vertices

(b) Loss graph on 10K iteration

Figure 18: **Computational analysis on operation time and loss convergence.** (a) The optimization time is plotted against input complexity (sequence length and mesh resolution), highlighting the scalability of our approach. (b) Training loss trajectories for MIXAMO and DT4D datasets confirm that the proposed framework ensures smooth and stable convergence within approximately 10K iterations.

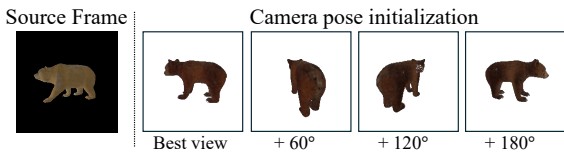

Figure 19: **Visualization of noisy camera initialization.** We illustrate the perturbed starting poses for optimization, created by adding $60°$, $120°$, and $180°$ of angular perturbation to the best view. These perturbations simulate imperfect calibration to evaluate the robustness of our pipeline.

Table 9: **Robustness to camera initialization.** Our method outperforms baselines with negligible degradation, even under $180°$ rotation.

| Method / Configuration | PMD ↓ | FID ↓ |
|---|---|---|
| Transfer4D | 0.0027 | 0.1535 |
| NPR$^+$ | 0.0136 | 0.1395 |
| Ours (Best Init.) | **0.0016** | **0.0193** |
| Ours ($+60°$ error) | 0.0018 | 0.0294 |
| Ours ($+120°$ error) | 0.0017 | 0.0259 |
| Ours ($+180°$ error) | **0.0016** | 0.0240 |

that the loss decreases smoothly across diverse assets, including both quadrupeds and humanoids, with convergence occurring at approximately 10K iterations. Across all evaluated scenarios, the end-to-end runtime stays well below 10 minutes on an RTX 4090.

### A.4.6   ANALYSIS ON CAMERA INITIALIZATION

To evaluate the robustness of our framework against imperfect camera initialization in the optimization stage, we intentionally introduce significant perturbations to the initial camera rotation. We test angular deviations of $60°$, $120°$, and $180°$ relative to the optimal initialization(Fig. 19), which is configured by render-and-compare (Sec. A.1).

As summarized in Tab. 9, our method exhibits remarkable stability even under extreme noise. Notably, initializing from a completely opposite viewpoint ($180°$ deviation) results in negligible performance degradation (PMD: $0.00159 \rightarrow 0.00160$), maintaining superiority over baseline methods.

Table 10: **Quantitative robustness analysis based on EMF.** We measure quality across different camera motion scenarios. EMF (angular) quantifies the effective angular coverage; lower values indicate limited parallax and increased geometric ambiguity.

| | EMF | Ours | | NPR$^+$ | |
|---|---|---|---|---|---|
| Scenario | (ang.) | PMD ↓ | FID ↓ | PMD ↓ | FID ↓ |
| Stationary | 35.22 | 0.0014 | 0.0075 | - | - |
| Slow orbit (30°) | 45.46 | 0.0011 | 0.0068 | 0.0053 | 0.0257 |
| Teleporting views | 253.8 | 0.0008 | 0.0083 | - | - |

Table 11: **Quantitative evaluation of multi-view motion transfer.** Results demonstrate consistent gains as the number of views increases (from 1 to 4). Notably, our method surpasses the NPR$^+$ baseline even in the challenging monocular setting.

| | Ours | | NPR$^+$ | |
|---|---|---|---|---|
| #Views | PMD ↓ | FID ↓ | PMD ↓ | FID ↓ |
| 1-View | 0.0027 | 0.0145 | 0.0045 | 0.0683 |
| 2-Views | 0.0022 | 0.0095 | - | - |
| 4-Views | 0.0020 | 0.0095 | - | - |

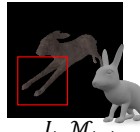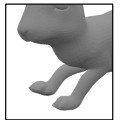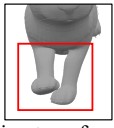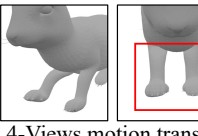

$I_t, \mathcal{M}_{tgt}$ | 1-View motion transfer | 4-Views motion transfer

Figure 20: **Qualitative comparison of multi-view integration.** We compare results from 1-view and 4-view inputs. The use of 4 views effectively resolves the inherent 2D-3D ambiguity observed in the single-view case, ensuring accurate geometric consistency.

Table 12: **Robustness to fast motion.** We simulate acceleration ($1.0\times$ to $3.0\times$) to induce motion blur. *Laplacian Variance* (LV) quantifies sharpness (lower values indicate severe blur).

| Speed (Frames) | LV | PMD ↓ | FID ↓ |
|---|---|---|---|
| 1.0× (66 frames) | 459.95 | 0.0010 | 0.0029 |
| 2.0× (33 frames) | 453.99 | 0.0010 | 0.0052 |
| 3.0× (22 frames) | 438.33 | 0.0011 | 0.0050 |

This robustness arises from our hierarchical optimization strategy, which decouples global alignment from local articulation. The root transformation is optimized first to quickly compensate for camera initialization errors, while internal motions are learned as relative transformations defined with respect to the canonical rest pose. This separation ensures that local pose optimization remains stable and unaffected by inaccuracies in the initial global orientation.

### A.4.7 ROBUSTNESS UNDER OCCLUSION

We assess robustness to occlusion using the Effective Multi-view Factor (EMF) from Gao et al. (2022), which measures the extent of viewpoint diversity in a monocular sequence. Low EMF indicates minimal camera motion, such as fixed or subtle hand-held captures, where severe self-occlusion and limited parallax make 3D reasoning highly ambiguous. In contrast, high EMF corresponds to larger viewpoint changes that provide multiple effective views, thereby reducing reconstruction ambiguity.

The quantitative results in Tab. 10 demonstrate that our method yields stable PMD and FID scores even in these low-EMF settings. Notably, in the stationary camera scenario, our method drastically reduces error compared to the baseline (PMD: 0.0014 vs. 0.0053), validating our effective handling of ambiguity without relying on large camera baselines.

### A.4.8 MULTI-VIEW SCALABILITY

We investigate whether introducing stronger multi-view cues alleviates inherent 2D-to-3D ambiguities. To this end, we evaluate performance variance across varying numbers of input viewpoints. As shown in Tab. 11, both PMD and FID metrics exhibit consistent improvement as the number of views increases from 1 to 4. This trend indicates that our model effectively exploits multi-view constraints to resolve 2D-to-3D ambiguities (Fig. 20). Notably, a distinct advantage of our framework is its scalability; the pipeline seamlessly extends to multi-view setups via differentiable rendering without requiring any modifications to the underlying network architecture.

### A.4.9 ROBUSTNESS AGAINST FAST MOTION AND MOTION BLUR

We evaluate robustness to rapid motion by rendering source videos at different playback speeds ($1\times$, $2\times$, $3\times$) with explicit motion blur in Blender (Blender). As shown in Tab. 12, our method re-

Table 13: **Quantitative evaluation across all scenes from DT4D-Quadrupeds.** Lower is better for both PMD and FID ($\downarrow$). Best and second-best results are highlighted in red and orange, respectively.

| Method | Punch PMD↓ | FID↓ | Walk1 PMD↓ | FID↓ | Death PMD↓ | FID↓ | Walk2 PMD↓ | FID↓ | KickBack PMD↓ | FID↓ |
|---|---|---|---|---|---|---|---|---|---|---|
| NPR[+] | 0.0027 | 0.0961 | 0.0027 | 0.1535 | 0.0039 | 0.0215 | 0.0010 | 0.0118 | 0.0024 | 0.0245 |
| Transfer4D | 0.0032 | 0.0145 | 0.0136 | 0.1395 | 0.0047 | 0.0399 | 0.0019 | 0.0099 | 0.0045 | 0.0383 |
| Ours | 0.0012 | 0.0074 | 0.0009 | 0.0029 | 0.0020 | 0.0343 | 0.0003 | 0.0043 | 0.0020 | 0.0211 |

| Method | Swim PMD↓ | FID↓ | Jump PMD↓ | FID↓ | Walk3 PMD↓ | FID↓ | Aggression PMD↓ | FID↓ | Howl PMD↓ | FID↓ |
|---|---|---|---|---|---|---|---|---|---|---|
| NPR[+] | 0.0030 | 0.0676 | 0.0024 | 0.0526 | 0.0022 | 0.0369 | 0.0025 | 0.0489 | 0.0022 | 0.0285 |
| Transfer4D | 0.0062 | 0.0837 | 0.0026 | 0.0055 | 0.0078 | 0.0388 | 0.0042 | 0.0075 | 0.0026 | 0.0079 |
| Ours | 0.0040 | 0.0385 | 0.0013 | 0.0085 | 0.0005 | 0.0023 | 0.0019 | 0.0335 | 0.0015 | 0.0141 |

| Method | Hit Back PMD↓ | FID↓ | Run Stop PMD↓ | FID↓ | Run Forward PMD↓ | FID↓ | Drink PMD↓ | FID↓ | Hop Forward PMD↓ | FID↓ |
|---|---|---|---|---|---|---|---|---|---|---|
| NPR[+] | 0.0013 | 0.0234 | 0.0059 | 0.3740 | 0.0083 | 0.0269 | 0.0043 | 0.0152 | 0.0026 | 0.0227 |
| Transfer4D | 0.0014 | 0.0434 | 0.0123 | 0.2292 | 0.0123 | 0.0303 | 0.0065 | 0.0207 | 0.0029 | 0.0485 |
| Ours | 0.0008 | 0.0077 | 0.0024 | 0.0377 | 0.0057 | 0.0261 | 0.0016 | 0.0134 | 0.0013 | 0.0050 |

Table 14: **Quantitative evaluation across all scenes from the Mixamo dataset.** Lower is better for both PMD and FID ($\downarrow$). Best and second-best results are highlighted in red and orange, respectively.

| Method | JumpingJacks PMD↓ | FID↓ | Running PMD↓ | FID↓ | SideStep PMD↓ | FID↓ | SkinningTest PMD↓ | FID↓ | StandingJump PMD↓ | FID↓ | SwingDance PMD↓ | FID↓ |
|---|---|---|---|---|---|---|---|---|---|---|---|---|
| SPT[+] | 0.0016 | 0.0047 | 0.0030 | 0.0069 | 0.0022 | 0.0170 | 0.0036 | 0.0376 | 0.0025 | 0.1816 | 0.0019 | 0.0143 |
| NPR[+] | 0.0017 | 0.0027 | 0.0287 | 0.0194 | 0.0042 | 0.0308 | 0.0092 | 0.0656 | 0.0084 | 0.2369 | 0.0029 | 0.0167 |
| Transfer4D | 0.0077 | 0.0107 | 0.0122 | 0.0450 | 0.0066 | 0.0294 | 0.0086 | 0.0664 | 0.0098 | 0.5631 | 0.0050 | 0.0108 |
| Ours | 0.0010 | 0.0035 | 0.0042 | 0.0229 | 0.0013 | 0.0089 | 0.0042 | 0.0195 | 0.0033 | 0.1728 | 0.0018 | 0.0087 |

| Method | Walking PMD↓ | FID↓ | Floating PMD↓ | FID↓ | HipHopDance PMD↓ | FID↓ | Header PMD↓ | FID↓ | Dying PMD↓ | FID↓ | Snatch PMD↓ | FID↓ |
|---|---|---|---|---|---|---|---|---|---|---|---|---|
| SPT[+] | 0.0018 | 0.0079 | 0.0074 | 0.0192 | 0.0027 | 0.0047 | 0.0036 | 0.0156 | 0.0025 | 0.0314 | 0.0024 | 0.0982 |
| NPR[+] | 0.0036 | 0.0057 | 0.0084 | 0.0059 | 0.0032 | 0.0028 | 0.0089 | 0.0177 | 0.0111 | 0.0772 | 0.0282 | 0.1798 |
| Transfer4D | 0.0113 | 0.0274 | 0.0043 | 0.0055 | 0.0034 | 0.0037 | 0.0071 | 0.0170 | 0.0072 | 0.0489 | 0.0173 | 0.1981 |
| Ours | 0.0011 | 0.0027 | 0.0037 | 0.0029 | 0.0032 | 0.0025 | 0.0026 | 0.0042 | 0.0043 | 0.0416 | 0.0029 | 0.0746 |

mains stable. Even with a reduction in Laplacian variance, indicating strong blur, the reconstruction metrics show minimal change (e.g., PMD: $0.0010 \rightarrow 0.0011$).

A known boundary case arises only when frames become fully degraded and contain no usable visual cues. In such cases, pose updates fail due to the absence of photometric or semantic gradients, a fundamental limitation of any image-supervised optimization method, rather than an issue specific to our approach.

## A.5 EXTENDED TABLES AND QUALITATIVE VIDEOS

**Quantitative evaluations across all scenes** We provide detailed quantitative results for all evaluation scenes from the DT4D (Li et al., 2021) and Mixamo (Adobe) datasets in Tab. 13, Tab. 14, and Tab. 15. These per-scene metrics supplement the averaged results presented in Tab. 1 of the main paper, consistently demonstrating our method's superior performance across diverse motion categories and scenarios.

**Qualitative Video Results** Qualitative comparisons between our approach and baseline methods are available via index.html file, or can be directly accessed in ./static/videos.

Table 15: **Quantitative evaluation across all scenes from the DT4D-others dataset.** Lower is better for both PMD (↓) and FID (↓). Best results are highlighted in **bold**. The DT4D-Others dataset contains animals that cannot be reconstructed with parametric templates, including birds, whales, dinosaurs, dragons, and elephants.

| Method | Fly | | Attack | | Running | | Walk | | Swimming | |
|---|---|---|---|---|---|---|---|---|---|---|
| | PMD ↓ | FID ↓ | PMD ↓ | FID ↓ | PMD ↓ | FID ↓ | PMD ↓ | FID ↓ | PMD ↓ | FID ↓ |
| Transfer4D | 0.0283 | 0.0971 | 0.0086 | 0.0409 | 0.0162 | 0.0165 | 0.0119 | 0.2455 | 0.0015 | **0.0024** |
| Ours | **0.0045** | **0.0415** | **0.0022** | **0.0122** | **0.0033** | **0.0028** | **0.0006** | **0.0020** | **0.0007** | 0.0034 |

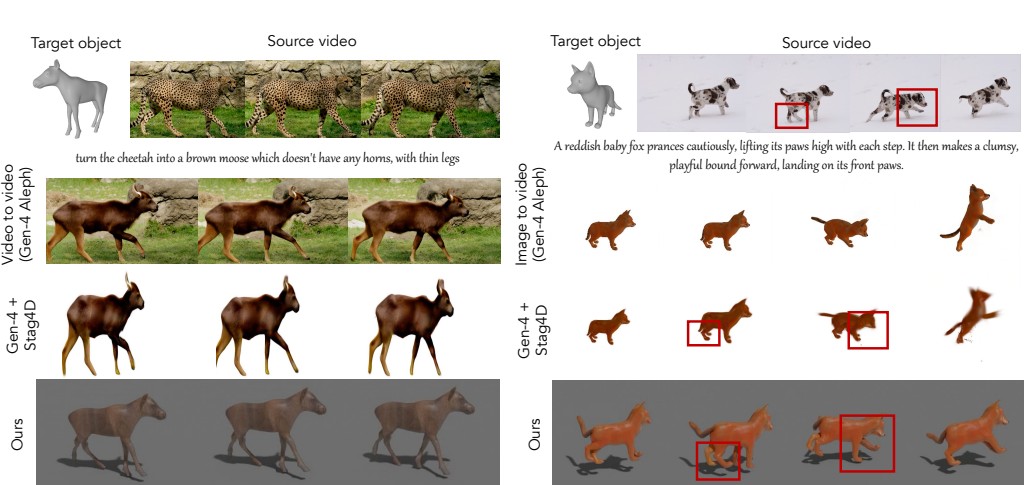

Figure 21: **Qualitative comparison with a generative 4D pipeline (Gen-4 Aleph (Runway) + Stag4D (Zeng et al., 2024)).** We illustrate the key limitations of generative approaches. *Left (Identity drift):* The generative pipeline does not fully preserve the geometry of the target object (moose mesh), leading to a more generic appearance even with appropriate text conditioning. *Right (Motion control specificity):* The highlighted regions (red boxes) show that the generative pipeline often fails to reconstruct the motion details.

## A.6 COMPARISON WITH GENERATIVE 4D PIPELINES

To situate our method within the broader landscape of generative video and 4D content creation, we compare against a representative state-of-the-art pipeline that integrates video-to-video generation (Runway Gen-4 (Runway)) with a 4D lifting approach (Stag4D (Zeng et al., 2024)).

We assess this pipeline under two practical usage configurations: (i) editing the source video by providing a rendered target image along with a text prompt describing the target's appearance (Fig. 21, left), and (ii) performing image-to-video generation using a single rendered view of the target while conditioning on a text prompt that specifies the source motion (Fig. 21, right). These settings correspond to common workflows in generative content production and serve as an appropriate reference for comparison.

Although this pipeline is capable of generating visually compelling results, its design goal differs fundamentally from our framework CAMO, which focuses on accurate motion retargeting for a specified target asset. The key distinctions are summarized below.

**Identity Preservation.** As shown on the left side of Fig. 21, text- or latent-conditioned video generation models generally lack mechanisms to maintain strict correspondence to a particular non-human 3D asset. Consequently, the output often drifts toward generic appearances rather than retaining the asset's original structure. CAMO avoids such drift by explicitly preserving the asset's topology and identity throughout optimization.

**Motion Faithfulness.** As illustrated on the right side of Fig. 21, video generation models produce visually plausible sequences but tend to default to generic motion patterns (e.g., walking, running, dancing). Such models are not designed to follow the precise temporal cues or the nuanced behaviors present in a specific driving video. In contrast, CAMO is designed to explicitly enforce temporal and geometric alignment, enabling the accurate transfer of fine-grained and idiosyncratic motion

characteristics. In addition, the 4D lifting stage introduces further temporal and geometric inconsistencies, as the generative video lacks the stable correspondence and motion specificity required for reliable reconstruction.

**Computational efficiency.** The two-stage generative pipeline is computationally demanding (approximately 1 hour per 120-frame sequence), whereas CAMO achieves significantly faster processing ( 10 minutes per 120 frames) while maintaining reliable control.

## B    THEORETICAL ANALYSIS

This section examines the structure of our morphology-parameterized representation. We aim to understand how the parameters that describe scale, bone lengths, surface offsets, and pose are constrained by an articulated object observed in motion. Under assumptions on piecewise rigidity, a kinematic tree, and sufficiently varied motion, we show that these proposed parameters are identifiable up to a single global scale factor. This explains why the formulation avoids unnecessary ambiguity and leads to stable optimization.

### B.1    PROBLEM SETUP AND ASSUMPTIONS

Let $\mathcal{I} = \{(I_t, M_t)\}_{t=0}^{T}$ denote the observed monocular sequence. We estimate two groups of unknowns: the time-varying pose parameters $\{\Theta_t\}$ and the time-invariant morphology parameters $\Phi$. The following analysis clarifies when these parameters are uniquely determined and how each component is fixed by the object's structure and observed motion.

**Pose Parameters ($\Theta$):**    The dynamic state of the articulated structure at time $t$:

$$\Theta = \{\Theta_t\}_{t=0}^{T} \tag{15}$$

where $\Theta_t$ includes the root global transformation and local joint rotations.

**Morphology Parameters ($\Phi$):**    The time-invariant parameters defining the target character's unique geometry:

$$\Phi = (s_{\text{global}}, \{l_b\}_{b \in \mathcal{B}}, \{o_i\}_{i=1}^{N}) \tag{16}$$

- $s_{\text{global}} \in \mathbb{R}^+$: Global scale factor to resolve monocular depth-scale ambiguity.
- $\{l_b\}_{b \in \mathcal{B}}$: Learnable lengths of bones $b$ in the kinematic tree $\mathcal{B}$.
- $\{o_i\}_{i=1}^{N}$: Local offsets for the $N$ 3D Gaussian primitives, modeling surface details.

The canonical center $\overline{\mu}_i$ of the $i$-th Gaussian is parameterized explicitly to couple the skeletal structure with the volumetric representation:

$$\overline{\mu}_i = s_{\text{global}}(p_i(l) + o_i) \tag{17}$$

Here, $p_i(l)$ is the skeleton-driven reference position determined by Linear Blend Skinning (LBS) weights $w_{ij}$ and the rest-pose joint locations $j_{\text{rest}}$:

$$p_i(l) = \sum_{j \in \mathcal{J}} w_{ij} j_{\text{rest}}(j; l) \tag{18}$$

The rest-pose joint positions are linear functions of the bone lengths:

$$\mathbf{j}_{\text{rest}}(j; l) = \mathbf{j}_{\text{rest}}(j_{root}) + \sum_{b \in P(root, j)} l_b v_b \tag{19}$$

where $v_b$ is the unit direction vector of bone $b$, and $P(root, j)$ denotes the set of parent joints along the kinematic chain from the root to joint $j$.

The optimization problem minimizes the energy function $E$:

$$(\Theta^*, \Phi^*) = \arg \min_{\Theta, \Phi} \left[ \mathcal{L}_{\text{render}} + \lambda_{\text{keypoint}} \mathcal{L}_{\text{keypoint}} + \lambda_{\text{reg}} \mathcal{L}_{\text{reg}} \right] \tag{20}$$

**Assumptions.**    To isolate the contribution of our morphology parameterization, we analyze identifiability under controlled conditions where ambiguities arising from 2D observations (e.g., occlusion, limited viewpoints) are ignored. Specifically, we rely on the following assumptions:

- **A1. Piecewise rigidity:** The target mesh consists of rigid parts connected by joints, forming a kinematic tree structure.

- **A2. Non-degenerate motion:** The motion observed in the source video exhibits sufficient rotation around linearly independent axes, avoiding planar or single-axis singularities that would prevent unique 3D structure recovery.

- **A3. Sufficient observability:** We assume that the temporal sequence of 2D observations provides sufficient viewpoint diversity (effectively serving as multi-view constraints) to resolve the 3D structure of rigid components.

**Remark.** In this theoretical analysis, we assume sufficient observability via photometric cues (A3) so that the identifiability of morphology parameters can be examined in isolation without confounding factors arising from incomplete or ambiguous 2D evidence. To resolve ambiguities under partial occlusion, we address this with dense semantic correspondences (Sec. 3.3), regularizations (Appendix A.2). Empirical robustness under limited viewpoint variation and occlusion is discussed in Appendix A.4.7– A.4.8.

### B.2 AMBIGUITY ANALYSIS OF THE NAIVE FORMULATION

Before introducing our morphology parameterization, we analyze a *naive model* to demonstrate why standard vertex-based optimization suffers from shape-pose entanglement under monocular supervision. In a naive formulation, while an articulated skeleton drives deformation via LBS, the canonical Gaussian centers (initialized from mesh vertices) are treated as free optimization variables without explicit morphological constraints relative to the skeleton (e.g., learnable bone lengths).

**Naive Gaussian-Center Model.** Let the canonical centers $\{\mu_i\}_{i=1}^N \in \mathbb{R}^3$ be directly optimized as independent variables. The rendering process at time $t$ from viewpoint $v$ is defined as:

$$\hat{I}_{t,v} = \Pi_v\big(\text{LBS}(\{\mu_i\}, \Theta_t)\big), \tag{21}$$

where $\Pi_v$ is the projection operator and $\Theta_t$ represents the articulated pose. Ideally, morphological discrepancies between the source and target should be absorbed exclusively by the static shape parameters (the canonical centers $\{\mu_i\}$), while pose parameters $\{\Theta_t\}$ solely capture the dynamic motion. However, we show that this disentanglement fails under 2D supervision alone.

**Proposition 1** (Shape–Pose Ambiguity in the Naive Model). *Let* $(\{\mu_i^*\}, \{\Theta_t^*\}_{t=0}^T)$ *be a solution that minimizes the reprojection error, where* $\{\mu_i^*\}$ *represents the time-invariant canonical shape. There exists a continuous family of alternative solutions* $(\{\tilde{\mu}_i\}, \{\tilde{\Theta}_t\})$ *that produce nearly identical rendered images.*

Specifically, for a perturbation in canonical shape $\{\Delta\mu_i\}$ that modifies the shape while preserving skeleton topology, there exists a corresponding pose adjustment $\{\Delta\Theta_t\}_{t=0}^T$ such that:

$$\forall t, \quad \left\| \frac{\partial \hat{I}_t}{\partial \mu}\{\Delta\mu_i\} + \frac{\partial \hat{I}_t}{\partial \Theta_t}\Delta\Theta_t \right\|_2 \approx 0. \tag{22}$$

**Why the Naive Model suffers Ambiguity.** The fundamental challenge lies in disentangling morphological adaptation from pose dynamics. Ideally, structural parameters (e.g., bone lengths) should adapt to the source's morphology while independently recovering the articulated *pose*. However, the naive formulation treats canonical Gaussians $\{\mu_i\}$ as free variables decoupled from the skeleton. This surface-skeleton decoupling allows the optimizer to satisfy projection constraints by incorrectly sliding surface points along bone axes rather than estimating the true pose. Consequently, this shape-pose ambiguity creates a degenerate solution space where morphological distortions erroneously compensate for pose estimation errors.

### B.3 IDENTIFIABILITY ANALYSIS

We now establish that our morphology parameterization alleviates the shape-pose entanglement. The key insight is that by explicitly coupling surface geometry to skeletal structure, we transform an underconstrained problem into one with unique solution.

### B.3.1 IDENTIFIABILITY THEOREM

**Theorem 1** (Identifiability under Morphology Parameterization). *If surface geometry is parameterized as:*

$$\mu_i(\Phi) = s_{global} \left( p_i(\{l_b\}) + o_i \right), \tag{23}$$

*where $p_i(\{l_b\})$ is the skeleton-driven joint position and $o_i$ is a local offset, then the morphology parameters $\Phi = (s_{global}, \{l_b\}, \{o_i\})$ and pose parameters $\Theta = \{\Theta_t\}_{t=0}^{T}$ are uniquely identifiable up to a global similarity transformation.*

*Proof.* We construct the solution through sequential decomposition, demonstrating that each parameter set is uniquely determined given the previous ones.

**Step 1: Rigid Part Decomposition.** By A1 (piecewise rigidity), the target object can be decomposed into $K$ rigid parts $\{\mathcal{P}_k\}_{k=1}^{K}$, each moving rigidly over time. Let $\tilde{\mu}_i^k$ denote the canonical (time-invariant) coordinates of point $i$ on part $\mathcal{P}_k$, and let $R_k(t) \in \mathrm{SO}(3)$ and $\mathbf{t}_k(t) \in \mathbb{R}^3$ denote the time-varying rigid transformation of part $\mathcal{P}_k$ at time $t$. Then the 3D trajectory of each point on $\mathcal{P}_k$ can be written as:

$$\mu_i^k(t) = R_k(t)(s_k \tilde{\mu}_i^k) + \mathbf{t}_k(t), \tag{24}$$

where $s_k > 0$ is an (unknown) isotropic scale associated with part $\mathcal{P}_k$.

Assumption A3 (sufficient observability) implies that the temporal observations provide enough independent constraints to uniquely determine the rigid motion of each part and its canonical shape up to this isotropic scale. In other words, for each $k$, the factorization

$$\{\mu_i^k(t)\}_{i,t} \quad \longleftrightarrow \quad \left( \{R_k(t), \mathbf{t}_k(t)\}_t, \{s_k \tilde{\mu}_i^k\}_i \right) \tag{25}$$

is unique up to the per-part scale $s_k$. Consequently, under non-degenerate motion (A2), we can recover the time-varying rigid transformations $\{R_k(t), \mathbf{t}_k(t)\}$ and the unscaled geometry for each part $k$ up to $s_k$.

**Step 2: Scale Unification via Kinematic Constraints.** Although Step 1 leaves an arbitrary local scale $s_k$ for each part, the kinematic tree imposes compatibility constraints at joints. Consider two adjacent parts $\mathcal{P}_k$ and $\mathcal{P}_{k'}$ connected at joint $j$. Let $\tilde{J}_j^k$ and $\tilde{J}_j^{k'}$ denote the corresponding joint locations in the canonical frames of $\mathcal{P}_k$ and $\mathcal{P}_{k'}$, respectively. Their world-space joint position at time $t$ must coincide:

$$R_k(t)(s_k \tilde{J}_j^k) + \mathbf{t}_k(t) \equiv R_{k'}(t)(s_{k'} \tilde{J}_j^{k'}) + \mathbf{t}_{k'}(t) \quad \forall t. \tag{26}$$

Rearranging equation 26 and with non-degenerate relative motions (A2) eliminates the translations and shows that the ratio:

$$\rho_{k \to k'} := \frac{s_{k'}}{s_k} \tag{27}$$

is uniquely determined by the recovered canonical geometries and motions. Intuitively, the physical bone incident to joint $j$ must have the same length when measured from either side, which fixes $s_{k'}/s_k$.

Because the kinematic graph is a tree, these ratios can be propagated from the root part $k = 0$ to all other parts:

$$s_k = \rho_{0 \to k} s_0, \tag{28}$$

where $\rho_{0 \to k}$ is determined by the unique path from the root to $k$. Thus all local scales $\{s_k\}$ become linear functions of a single global scale $s_0$. This reduces the scale degrees of freedom from $K$ (independent per-part scales) to a single scalar $s_{\text{global}} \equiv s_0$.

**Step 3: Bone Lengths and Pose.** With a unified global scale $s_{\text{global}}$, the joint trajectories $\{J_j(t)\}$ are uniquely determined in world coordinates. For a bone $b$ connecting a parent joint $j_{\text{parent}}$ and a child joint $j_{\text{child}}$, its physical length is:

$$\ell_b \;=\; \|J_{j_{\text{child}}}(t) - J_{j_{\text{parent}}}(t)\|, \tag{29}$$

which is invariant over time. Hence the set of bone lengths $\{\ell_b\}$ is uniquely determined (up to the same global scale already absorbed into $s_{\text{global}}$).

Given the kinematic tree, the known bone lengths $\{\ell_b\}$, and the joint trajectories $\{J_j(t)\}$, the pose parameters $\{\Theta_t\}$ (e.g., joint rotations in a chosen parameterization) are obtained by solving the inverse kinematics (IK) problem at each time $t$. Assumption A2 (non-degenerate motion with sufficient rotational variation) ensures that the IK solution is unique, i.e., discrete ambiguities such as mirrored configurations are ruled out by temporal continuity and multi-joint consistency. Therefore, the pose sequence $\{\Theta_t\}$ is uniquely determined.

**Step 4: Surface Offsets.** Finally, we consider the morphology parameters associated with the surface, namely the local offsets $\{o_i\}$. As defined in Sec. 3.2, each canonical Gaussian mean is parameterized as:

$$\bar{\mu}_i \;=\; s_{\text{global}}\big(p_i(\{\ell_b\}) + o_i\big), \tag{30}$$

where $p_i(\{\ell_b\})$ is the skeleton-anchored reference position obtained from the kinematic chain and skinning weights, and $o_i$ is a time-invariant offset in the canonical frame.

Given the recovered pose sequence $\{\Theta_t\}$, the LBS operator deterministically maps canonical positions to their deformed positions. Thus each observed deformed point $\tilde{\mu}_i(t)$ satisfies:

$$\tilde{\mu}_i(t) \;=\; \text{LBS}\big(s_{\text{global}}\big(p_i(\{\ell_b\}) + o_i\big), \; \Theta_t\big). \tag{31}$$

Here, $s_{\text{global}}$, $\{\ell_b\}$, and $\{\Theta_t\}$ are already fixed by Steps 2–3, so equation 31 is linear in $o_i$ for each time $t$. Stacking equation 31 over multiple time steps yields an overdetermined linear system for $o_i$. By A2, the poses $\{\Theta_t\}$ span sufficiently diverse configurations so that the corresponding system has full column rank, and thus admits a unique least-squares solution for $o_i$.

Putting all steps together, we conclude that under A1–A3, the morphology parameters $(s_{\text{global}}, \ell_b, o_i)$ and the pose sequence $\{\Theta_t\}$ are uniquely determined up to a single global scale factor $s_{\text{global}}$, completing the identifiability. $\qquad\square$

**Remark.** This sequential identifiability theoretically justifies our hierarchical optimization strategy. By prioritizing global structure (scale, bone lengths) before refining local offsets, we align the optimization trajectory with the identifiable path derived above, ensuring stable convergence.

### B.3.2 OPTIMIZATION ANALYSIS

In this subsection, we study how each parameter group in $\Psi = \{s_{\text{global}}, \ell_b, \Theta, o_i\}$ influences the rendered image by examining their induced image-space motion fields. Rather than evaluating the full rendering Jacobian $J = \partial\hat{I}/\partial\Psi$ directly, we analyze the instantaneous 2D displacement (or gradient flow) generated by perturbing each parameter. This provides geometric intuition for how CAMO achieves disentangled and stable optimization.

We focus on the geometric component of the rasterization process by only considering the projection of 3D Gaussian centers. Let $\mu \in \mathbb{R}^3$ be the mean position of a 3D Gaussian and $u = \pi(\mu) \in \mathbb{R}^2$ be its perspective projection on the image plane. Using the chain rule, the 2D motion field $F_\psi(u)$ induced by a parameter $\psi$ is formulated as:

$$F_\psi(u) = \frac{\partial\pi}{\partial\mu}\frac{\partial\mu}{\partial\psi} = \mathbf{J}_\pi(\mu) \cdot \mathbf{v}_\psi, \tag{32}$$

where $\mathbf{J}_\pi(\mu) \in \mathbb{R}^{2\times3}$ is the Jacobian of the perspective projection function at $\mu$, and $\mathbf{v}_\psi \in \mathbb{R}^3$ is the instantaneous 3D velocity of the Gaussian center induced by perturbing $\psi$. We analyze the structure of $\mathbf{v}_\psi$ and its projection for each parameter group:

**Global Scale** $s_{global}$**.** Scaling uniformly moves points along the ray from the camera origin. The induced 3D velocity is radial, $\mathbf{v}_s \propto \mu$. Under perspective projection, this results in a purely radial motion field centered at the principal point $c$:

$$F_{s_{global}}(u) \propto (u - c). \tag{33}$$

This creates a global, low-frequency expansion/contraction pattern.

**Bone Length** $\ell_b$**.** Let $w_{ib}$ be the LBS skinning weight of the Gaussian $i$ with respect to bone $b$. Varying the bone length shifts child Gaussians along the bone axis vector $\mathbf{b}_{axis}$. The induced 3D velocity is $\mathbf{v}_b = w_{ib} \cdot \mathbf{b}_{axis}$. The projected motion field is:

$$F_{\ell_b}(u) = \mathbf{J}_\pi(\mu) \cdot (w_{ib} \cdot \mathbf{b}_{axis}). \tag{34}$$

Unlike global scale, this field is spatially localized to the specific limb and constrained to align with the vanishing point of the bone axis.

**Pose Parameters** $\Theta$**.** A pose update corresponds to a rigid rotation of a body part around a joint. Let $\omega$ be the instantaneous angular velocity vector derived from $\Theta$, and $p$ be the joint location. The induced 3D velocity is tangential to the arc of rotation: $\mathbf{v}_\Theta = \omega \times (\mu - p)$. The projected motion field captures the perspective projection of this arc:

$$F_\Theta(u) = \mathbf{J}_\pi(\mu) \cdot (\omega \times (\mu - p)). \tag{35}$$

Depending on the rotation axis relative to the view direction, this produces distinct curvilinear flow patterns (e.g., circular motion or foreshortening effects). These patterns are geometrically distinguishable from the strictly linear shifts caused by bone scaling.

**Local Offsets** $o_i$**.** Offsets model fine-grained surface details independent of the skeletal structure. A perturbation in $o_i$ induces an arbitrary local 3D displacement $\mathbf{v}_{o_i}$.

$$F_{o_i}(u) = \mathbf{J}_\pi(\mu) \cdot \mathbf{v}_{o_i}. \tag{36}$$

Crucially, because $o_i$ operates on individual Gaussians rather than kinematic chains, it generates high-frequency, sparse motion updates. This sparsity makes the offset gradients orthogonal to the global, low-frequency motion fields induced by scale, bone length, and pose, preventing optimization ambiguity.

