# OpenReview forum: "CAMO: Category-Agnostic 3D Motion Transfer from Monocular 2D Videos"
_ICLR.cc/2026/Conference — Submitted to ICLR 2026_

### Official Review · Reviewer_S7f1 · 2025-10-27

**Soundness:** 2
**Presentation:** 3
**Contribution:** 2
**Rating:** 4
**Confidence:** 4

**Summary:**

The paper introduces CAMO, a category-agnostic 3D motion transfer framework that maps motions from a monocular 2D video onto arbitrary 3D targets—without using category templates or reconstructing source 3D meshes. CAMO represents the target as an articulated 3D Gaussian Splatting model driven by an LBS-based kinematic chain. It learns morphology-adaptive parameters (bone lengths, scale, offsets) to handle shape differences and uses dense 2D–3D semantic correspondences to reduce pose ambiguity. The total loss combines differentiable rendering, keypoint, and regularization terms. Experiments on DT4D and Mixamo show state-of-the-art performance in motion accuracy and visual realism, outperforming template-based and reconstruction-to-retarget baselines, while maintaining fast optimization (<10 min per sequence).

**Strengths:**

1. Category-agnostic design: Avoids SMPL/SMAL or per-class priors; directly optimizes on the target with only monocular source supervision. This reduces error cascade from reconstruct→retarget pipelines.
2. Clarity: The paper is well-presented, easy to follow.
3. Clear empirical gains: SOTA PMD/FID across datasets; ablations isolate the contributions of rendering loss, shape param., and keypoints; efficient optimization on commodity hardware.

**Weaknesses:**

1. While image-space supervision is attractive, robustness under heavy occlusion or fast motion isn’t systematically quantified; correspondence quality in such regimes is unclear.
2. Real-world setup relies on a render-and-compare camera initialization. The method’s sensitivity to poor initial camera guesses or to calibration drift is not analyzed.
3. For experiments, the authors should provide more visualization results. The demo's examples can not convince me for robust results.
4. About comparison, I'm curious about the comparison with a simple pipeline: using video generation/editing model to get edited object, and then apply dynamic 3D generation model for animation. Can CAMO outperform it in all evaluation aspects?

**Questions:**

Please see weakness. If all my concerns are well conducted, I'll consider raise my score.

---

> ### Author Response · Authors · 2025-11-22
> **Response to reviewer S7f1 (1)**
>
> We sincerely thank the reviewer S7f1 for the insightful feedback and the opportunity to address your concerns. In response to your valuable suggestions, we have conducted additional experiments and added comprehensive materials. We address each concern in detail below.
>
> ---
>
> **W1. Regarding robustness.** To explicitly evaluate robustness under heavy occlusion and fast motion, which are the regimes mentioned in the comment, we conducted three targeted quantitative tests: (1) EMF-based occlusion analysis, (2) multi-view scenario tests, and (3) fast-motion and motion-blur stress tests.
>
> **(1) Robustness under Low-EMF Occlusion (Table A).** Because monocular sequences naturally vary in how much geometric information (i.e., parallax) the camera motion provides, we adopt the Effective Multi-view Factor (EMF) [1] to measure the *degree of view diversity* within a single monocular capture. For clarity, EMF can be interpreted as follows:
>
> - Low EMF: the camera moves very little (e.g., fixed or subtle hand-held motion). These settings yield stronger occlusion ambiguity, since the view is nearly constant.
> - High EMF: large viewpoint changes give multiple effective views, reducing ambiguity.
>
> We report the results across varying EMF values in the table below. As expected, higher EMF (stronger viewpoint diversity) gives the lowest PMD. Importantly, even when EMF becomes very low and the setup approaches a fixed, occlusion-heavy configuration, CAMO still maintains stable accuracy and continues to outperform all baselines by a clear margin.
>
> **[Table A] Qauntitative evaluations on monocular videos with varying EMF.**
>
> | Camera viewpoint | EMF (angular) | PMD(↓) | FID(↓) |
> | --- | --- | --- | --- |
> | Stationary | 35.2 | 0.0014  | 0.0075  |
> | Slow Orbit ($30^{\circ}$) | 45.5 | 0.0011 (NPR: 0.0053) | 0.0068 (NPR: 0.0257) |
> | Teleporting Views | 253.8 | 0.0008 | 0.0083 |
>
> **(2) Effect of additional views.** To directly test whether correspondence ambiguity (e.g., left–right confusion) decreases with stronger geometric cues, we increase the number of input viewpoints. As shown in Table B, both PMD and FID improve consistently from 1 → 2 → 4 views, indicating that the model effectively exploits additional viewpoints to resolve ambiguities. Importantly, CAMO seamlessly extends to multi-view inputs directly through differentiable rendering without requiring any architectural changes.
>
> **[Table B] Quantitative evaluations on multi-view motion transfer.**
>
> |  | PMD(↓) | FID(↓) |
> | --- | --- | --- |
> | 1 view | 0.0027 (NPR: 0.0045) | 0.0145 (NPR: 0.0683) |
> | 2 views | 0.0022 | 0.0095 |
> | 4 views | 0.0020 | 0.0095 |
>
> **(3) Robustness under fast motion and motion blur.** Finally, we simulated increasingly fast motion by generating the source video (1×, 2×, 3× speed) with explicit motion-blurred frames using Blender. As summarized in Table C, even though the Laplacian variance decreases (indicating stronger blur), PMD and FID remain stable (e.g., PMD: 0.0010 → 0.0011).
>
> However, we’d like to clearly mention that when intermediate frames are fully blurred and contain no usable visual cues, the optimization receives almost no meaningful photometric or semantic gradient, and therefore cannot reliably update the pose. This is a natural limitation of the image-supervised optimization framework.
>
> **[Table C] Quantitative evaluations on motion transfer under motion blur.**
>
> |  | average laplacian variance | PMD(↓) | FID(↓) |
> | --- | --- | --- | --- |
> | 1.0 (66 frames) | 459.95 | 0.0010 | 0.0029 |
> | 2.0 (33 frames) | 453.99 | 0.0010 | 0.0052 |
> | 3.0 (22 frames) | 438.33 | 0.0011 | 0.0050 |
>
> Overall, these evaluations clarify how CAMO behaves across low-EMF occlusion, multi-view, and fast-motion regimes.

---

> ### Author Response · Authors · 2025-11-22
> **Response to reviewer S7f1 (2)**
>
> **W2. About camera initialization during optimization.** Thank you for raising this practical point. To evaluate the sensitivity of our method to imperfect camera initialization, we perturb the initial camera rotation by large angles (60°, 120°, 180°) relative to the render-and-compare best initialization.
>
> As shown in Table D, CAMO remains stable even under extremely noisy initialization. For example, shifting the camera by 180° (opposite viewpoint) results in only a slight degradation in performance (PMD: 0.00159 → 0.00160, FID: 0.0193 → 0.0240), and still outperforms baselines.
>
> This robustness stems from our hierarchical optimization that decouples global alignment from local articulation. The root transformation is optimized to rapidly absorb camera initialization errors, while internal motions are learned as relative transformations from the canonical rest pose. This design ensures that local pose learning remains stable and independent of the initial global orientation.
>
> We have also added this sensitivity analysis to Appendix A.4.6.
>
> **[Table D] Qualitative evaluations on varying camera initialization.**
>
> |  | PMD(↓) | FID(↓) |
> | --- | --- | --- |
> | Transfer4D | 0.00269 | 0.1535 |
> | NPR | 0.01362 | 0.1395 |
> | best cam | 0.00159 | 0.0193 |
> |60° perturbation| 0.00175 | 0.0294 |
> |120° perturbation | 0.00165 | 0.0259 |
> |180° perturbation| 0.00160 | 0.0240 |
>
> **W3. More visualization results.** We appreciate the reviewer’s request for more comprehensive visual evidence. In response, we have expanded the supplementary materials with a broader set of qualitative results and an extended video, including:
>
> - 12 additional sequences from DeformingThings4D (DT4D) and MIXAMO, covering diverse articulated and deforming motions, (00:30~01:20)
> - 3 additional real-world motion-transfer examples across different object categories, (01:21~ 01:57)
> - side-by-side comparisons with baselines in real-world settings. (01:58~02:08)
>
> We believe these additions provide a more complete view of the method’s behavior and robustness across a wide range of scenarios.
>
> **W4. Comparison with Generative Pipeline (Video Gen + 4D Gen).** We thank the reviewer for this insightful suggestion and we'd like to refer to the supplementary video (01:58–02:08) and Figure 21 for this visual comparison. To investigate this setting, we deployed a motion transfer pipeline with the combination of state-of-the-art generative models (Gen-4 [2] + Stag4D [3]) and performed an analysis of the fundamental trade-offs.
>
> We utilized the Gen-4 [2] video generative model to synthesize edited videos. Subsequently, we applied Stag4D[3], which integrates pre-trained diffusion models with dynamic 3D Gaussian splatting, to achieve high-fidelity 4D generation. Please refer to Appendix A.6. for implementation details and more detailed discussions.
>
> While such pipelines produce visually plausible animations, their objective differs from that of CAMO, which aims for precise motion retargeting of a given asset. Below, we summarize the key differences and the resulting trade-offs.
>
> 1. *Identity Preservation*: Video generation models are primarily conditioned through text or latent features and struggle to enforce strict adherence to a specific non-human 3D asset. As a result, the synthesized subjects frequently exhibit identity drift, deviating toward generic or approximate shapes and appearances. CAMO, by contrast, is explicitly designed to preserve the asset’s topology and identity throughout the optimization process.
> 2. *Motion Faithfulness*: The generative pipeline tends to produce plausible but generic motions, often disregarding the fine-grained temporal structure and idiosyncratic motion cues present in the driving video. CAMO enforces sequential and geometric consistency, enabling the accurate transfer of such nuanced behavioral characteristics.
> 3. *Efficiency*: The generative pipeline is computationally expensive ($>$1 hour/120 frames), whereas CAMO offers efficient control ($<$10 mins/120 frames).
>
>  Although the generative pipeline can create visually appealing animations, it is not optimized for identity- and motion-preserving retargeting. CAMO consistently outperforms this baseline in these evaluation dimensions.
>
> We hope our clarifications address the reviewer’s question, and we would be happy to answer any further questions.
>
> ---
>
> References
>
> [1] Gao, Hang, et al. "Monocular dynamic view synthesis: A reality check." *Advances in Neural Information Processing Systems* 35 (2022): 33768-33780.
>
> [2] https://runwayml.com/
>
> [3] Zeng, Yifei, et al. "Stag4d: Spatial-temporal anchored generative 4d gaussians." *European Conference on Computer Vision*. Cham: Springer Nature Switzerland, 2024.

---

> ### Author Response · Authors · 2025-11-27
>
> Dear reviewer S7f1,
>
> Thank you again for taking the time to review our work.
>
> We would like to kindly check whether any additional clarification or discussion from our side would be helpful, particularly regarding the points addressed in our recent response.
>
> For your convenience, we summarize the main points and indicate where they are addressed in the revised manuscript below, marked in blue for clarity:
>
> - Regarding robustness (Appendix A.4.7-4.9, Table 10-12, Figure 20)
> - Camera initialization (Appendix A.4.6, Table 9, Figure 19)
> - More visual videos (Additional videos link: https://drive.google.com/file/d/1tR3ZgRONpi1gE5cwupXrQh7G8957f_2V/view?usp=sharing)
> - Comparison with generative pipeline (Appendix A.6, Figure 21)
>
> Please feel free to let us know if any further details would be useful.
>
> We sincerely appreciate your time and consideration during the review process.

---

### Official Review · Reviewer_gp2U · 2025-10-31

**Soundness:** 3
**Presentation:** 3
**Contribution:** 3
**Rating:** 6
**Confidence:** 3

**Summary:**

The paper proposes CAMO, a template-free, category-agnostic 2D→3D motion transfer method. It avoids reconstruct-then-retarget pipelines by optimizing the target asset directly in image space using (i) an articulated 3D Gaussian representation with morphology-adaptive parameters (bone lengths, global scale, local Gaussian offsets) and (ii) dense 2D to 3D semantic correspondences for disambiguation. Results show lower PMD/FID vs composite baselines (SPT+, NPR+) and Transfer4D across Mixamo & DT4D, plus qualitative real-world demos.

**Strengths:**

Sound and principled design.
The optimization objective (photometric + SSIM + semantic + temporal regularization) is coherent and mathematically well-founded.
The choice of directly optimizing in image space with an articulated 3D Gaussian structure is both elegant and technically solid.

Clear handling of 3D lifting.
The method performs implicit 3D lifting by analysis-by-synthesis, guided by differentiable rendering and dense 2D to 3D correspondences.
This eliminates explicit 3D supervision and is novel for the community.

Thorough ablations and visualizations.
The paper includes extensive quantitative and qualitative analyses (Tables 1–6, Figs. 5–14) demonstrating the necessity of morphology parameters and semantic losses. Failure cases and challenging cases are all presented and analyzed well.

**Weaknesses:**

1. Dependence on rigging quality and pre-processing.
The approach assumes access to well-rigged target meshes. Auto-rigging tools (e.g., UniRig, MagicArticulate) introduce noticeable artifacts when bone topology mismatches occur. Is there any way to address this?

2. Temporal scalability and long-sequence degradation.
The time-conditioned MLP cannot effectively model long (>600 frame) motion sequences; it gradually drifts or repeats poses. It's better to give some visualised results.

3. Absence of physical realism metrics.
Evaluation focuses on FID and PMD, but omits motion stability or contact-based metrics (e.g., foot-skating, interpenetration). Without such analysis, claims of realistic motion transfer remain visually but not physically validated.

4. Computation and convergence analysis are limited.
Although the authors report that optimization takes <10 minutes per sequence on an RTX 4090, there is no detailed runtime or convergence study across different mesh complexities or sequence lengths.

**Questions:**

CAMO delivers a robust, well-implemented, and innovative approach to category-agnostic 2D to 3D motion transfer.
The technical contribution and experimental validation are solid and can be a good contribution to the community.
The paper meets ICLR’s bar for novelty and soundness and is likely to stimulate follow-up work in differentiable 3D motion learning.
I strongly suggest that the authors open-source the source for reproducibility.

---

> ### Author Response · Authors · 2025-11-22
> **Response to reviewer gp2U (1)**
>
> We sincerely thank reviewer gp2U for the thorough and constructive feedback. We appreciate your recognition of our sound design, clear presentation, and solid experimental validation. We address each point in detail below and will release our code and assets to support reproducibility.
>
> **W1. Dependence on Rigging Quality and Pre-processing.** We appreciate the reviewer raising this concern. Topology mismatch in auto-rigging (e.g., missing or incorrectly formed joints) is a known limitation shared by skeleton-driven motion-transfer methods, and severe failure can indeed make accurate motion retargeting difficult. However, based on our empirical observations, CAMO remains robust across different automatic rigging pipelines. We evaluate the method using three widely used tools (RigNet, MagicArticulate, and UniRig), and observe only small performance variations. As shown in Table A, PMD and FID remain within a narrow range (e.g., PMD 0.0015 to 0.0018 and FID 0.0094 to 0.0153 on DT4D-sub), indicating that our framework compensates effectively for typical inaccuracies introduced during auto-rigging.
>
> **Table A. Ablation study on rigging modules.**
>
> |  | DT4D-sub (PMD↓) | DT4D-sub (FID↓) | DT4D-all (PMD↓) | DT4D-all (FID↓) |
> | --- | --- | --- | --- | --- |
> | CAMO+RigNet | 0.0018 | 0.0153 | 0.0019 | 0.0159 |
> | CAMO+MagicArticulate | 0.0016 | 0.0094 | 0.0021 | 0.0117 |
> | CAMO+UniRig | 0.0015 | 0.0110 | 0.0026 | 0.0168 |
>
> Our morphology-adaptive parameters (bone lengths and Gaussian offsets) help absorb moderate rigging inconsistencies, explaining the stability across different rigs. Severe topology failures (e.g., missing limbs or incompatible hierarchies) remain challenging because they provide insufficient structural cues for reliable articulation. Recent advances in template-free autoregressive rigging, such as RigAnything (SIGGRAPH TOG 2025) [1], are making automatic rigging increasingly robust across diverse and out-of-distribution assets, and integrating such tools is a promising way to further reduce dependence on rig quality.
>
> **W2. Temporal scalability and long-sequence visualized results.** We thank the reviewer for the suggestion. As requested, we added qualitative visualizations of long-sequence behavior in Appendix A.4.3. (Figure 15). Furthermore, we provide experimental results on scalability to longer frames below.
>
> While we observe that our time-conditioned MLP may degrade to more than 600 frames with the default sinusoidal time embedding (with frequency band 6), increasing the frequency of the sinusoidal embedding alleviates this issue and restores motion fidelity for longer sequences. Table B demonstrates that raising the frequency band from 6 to 8 improves performance from PMD 0.0012 → 0.0010 and FID 0.01251 → 0.00475, demonstrating that higher-frequency temporal encoding maintains discriminability for longer frames.
>
> **Table B. Qualitative evaluations on varying video lengths.**
>
> | # Frames / PE resolution | PMD (↓) | FID (↓) |
> | --- | --- | --- |
> | 100 / 6 | 0.0010 | 0.00342 |
> | 330 / 6 | 0.0008 | 0.00598 |
> | 660 / 6 | 0.0012 | 0.01251 |
> | 660 / 8 | 0.0010 | 0.00475 |
>
> For substantially longer sequences, our method offers flexible scalability via: (i) increasing the MLP capacity, or (ii) temporal segmentation, where the sequence is partitioned into segments with separate motion-field MLPs. This segmentation approach effectively mitigates drift and supports optimization stability over long durations. Given our framework’s high efficiency, the computational cost of optimizing multiple segments remains manageable, making it a practical solution for extended videos.
>
> ---
>
> References
>
> [1] Liu, Isabella, et al. "Riganything: Template-free autoregressive rigging for diverse 3d assets." *ACM Transactions on Graphics (TOG).*

---

> ### Author Response · Authors · 2025-11-22
> **Response to reviewer gp2U (2)**
>
> **W3. Physical Realism.**
> While our framework supports category-agnostic motion transfer, both our method and the baselines are currently designed primarily focusing on visual motion transfer rather than physical simulation. However, we strongly agree that incorporating physical realism would further enhance the practical applicability of our method.
>
> In particular, a natural next step is to augment our optimization with Jacobian-based physical constraints [1] and contact-aware regularization [2], which have been used to improve motion plausibility in both character animation and visuomotor control. These techniques can provide important kinematic consistency and contact stability directly from motion cues.
>
> Adding such constraints would allow our framework to penalize physically implausible motions, enforce contact stability, and integrate scene-dependent constraints. We included this discussion as an important direction for future work in the paper, and we believe the articulated Gaussian representation provides a suitable foundation for integrating such physically grounded priors into motion transfer.
>
> ---
>
> **W4. Computational efficiency and convergence.** We thank the reviewer for this suggestion. In the revised manuscript, we have included a comprehensive evaluation of computational costs in Appendix A.4.5. Below, we summarize the key experimental observations regarding runtime stability and convergence behavior.
>
> **Runtime Scalability.** To empirically validate the scalability of our method, we analyze the optimization time across the varying sequence lengths and mesh resolutions. We categorize the dataset into qualtiles (Q1-Q4) based on frame counts and vertex counts to observe performance trends across different complexity levels.
> As in Tables C below (and visualized in Figure 18 (a) of the revised paper), our optimization time remain stable (below 5 minutes to converge) as the frame number increases. In addition, the number of vertices only slightly increases the optimization time from 4K vertices (avg. 4.34 mins) up to 25K vertices (avg. 4.49 mins). Our hierarchical parameterization effectively decouples global structure from local geometry. This ensures that increasing mesh resolution restricts complexity to local offset updates without affecting global pose convergence, enabling efficient scaling with fairly restrained overhead.
>
> **Table C. Optimization time in accordance with the number of frames.**
>
> |Frame-bin|Frame-range|Time (min)|
> |-|-|-|
> |Q1|14–24|4.13|
> |Q2|25–33|4.41|
> |Q3|34–62|4.39|
> |Q4|67–140|4.60|
>
> **Table D. Optimization time in accordance with the number of vertices.**
>
> |Vertex-bin|Vertex-range|Time (min)|
> |-|-|-|
> |Q1|4K–10K|4.34|
> |Q2|10K–12K|4.34|
> |Q3|12K–15K|4.33|
> |Q4|15K–25K|4.49|
>
>
> **Convergence.** Empirically, our method demonstrates highly stable training dynamics across diverse morphological categories. For both quadruped and humanoid subjects, the optimization loss decreases smoothly and monotonically, typically stabilizing around 10k iterations. This stable behavior ensures that the entire process completes in under 5 minutes in most cases on a single RTX 4090. Detailed loss landscapes supporting these observations are visualized in Figure 18 (b) (Appendix A.4.5). Our evaluation datasets cover sequences with an average length of 46 frames (max 140 frames), and we have updated the manuscript to include these statistics.
>
> We hope our clarifications address the reviewer’s concerns, and we would be happy to discuss any further questions.
>
> ---
>
> References
>
> [1] Li, Sizhe Lester, et al. "Controlling diverse robots by inferring Jacobian fields with deep networks." *Nature* (2025).
>
> [2] Villegas, Ruben, et al. "Contact-aware retargeting of skinned motion." ICCV (2021).

---

> ### Author Response · Authors · 2025-11-27
>
> Dear reviewer gp2U,
>
> Thank you again for taking the time to review our work.
>
> We would like to kindly check whether any additional clarification or discussion from our side would be helpful, particularly regarding the points addressed in our recent response.
>
> For your convenience, we summarize the main points and indicate where they are addressed in the revised manuscript below, marked in blue for clarity:
>
> - Dependence on Rigging (Appendix A.3.3, Table 5)
> - Temporal Scalability (Appendix A.4.3, Table 8, Figure 14, 15)
> - Limitations and future work (Sec. 5)
> - Computational Analysis (Appendix A.4.5, Figure 18)
>
> Please feel free to let us know if any further details would be useful.
>
> We sincerely appreciate your time and consideration during the review process.

---

### Official Review · Reviewer_2Hny · 2025-11-01

**Soundness:** 2
**Presentation:** 2
**Contribution:** 2
**Rating:** 6
**Confidence:** 4

**Summary:**

This paper presents a category agnostic method for transferring motion from a 2d monocular video to a 3d target mesh. the method optimizes the pose of a target 3D model, represented by an articulated 3dgs framework, directly in the 2d observation space. The core components are a morphology-adaptive shape parameterization，which includes learnable bone lengths, global scale, and local offsets，and a dense semantic correspondence loss. This correspondence, derived from a pretrained feature extractor, aligns the 3d target with the 2d source.

**Strengths:**

1. The paper is well-written, structured, and easy to comprehend.
2. The quality of the results look good.
3. The methods handle category agnostic motion transfer without parametric models like SMPL and SMAL.
4. The use of articulated 3DGS for differentiable rendering , the leveraging of foundation models for robust semantic understanding , and the carefully designed morphology parameterization come together to form a reasonable and coherent framework.

**Weaknesses:**

1. Motion is parameterized by an MLP conditioned on a sinusoidal time embedding (Appendix A.2). This architecture is known to struggle with representing very long, complex, or non-cyclic motions, which is confirmed by the paper's own analysis (Appendix A.4) showing degradation on longer sequences.
2. The method is still vulnerable to fundamental 2D-to-3D ambiguities. The paper's failure cases (Fig. 14) show it can confuse left/right limbs or fail during severe self-occlusion, problems that persist despite the semantic correspondence loss.
3. Missing references, there are some works with similar settings:
[1] PhysRig: Differentiable Physics-Based Skinning and Rigging Framework for Realistic Articulated Object Modeling
[2] Puppeteer: Rig and Animate Your 3D Models

**Questions:**

1. Have you analyzed when these ambiguities are most likely to occur? For instance, does the left-right confusion happen most often from specific camera angles where the 2D projection is maximally ambiguous?

---

> ### Author Response · Authors · 2025-11-22
> **Response to reviewer 2Hny**
>
> We sincerely thank the reviewer 2Hny for the thoughtful and constructive feedback. We address each point in detail below.
>
> **W1. Temporal scalability.** We thank the reviewer for raising this point. As noted, our method naturally shares the known limitations of time-conditioned MLPs for very long or non-cyclic motions. We provide an detailed empirical study in Appendix A.4.3 analyzing this behavior on extended sequences.
>
> In our experiments, we observed that with the default sinusoidal time embedding, performance begins to degrade beyond approximately 600 frames. As the temporal domain expands, the sinusoidal embedding offers reduced temporal resolution, making the MLP input less discriminative for rapid or high-frequency motion changes.
>
> Consistent with this interpretation, increasing the frequency of the sinusoidal embedding alleviates this issue and restores motion fidelity for longer sequences. As shown in Table A, both the third and fourth rows evaluate a 660-frame sequence with different frequency bands. Raising the frequency band from 6 to 8 improves performance (PMD 0.0012 → 0.0010 and FID 0.01251 → 0.00475), indicating that higher-frequency temporal encoding better preserves motion fidelity in longer sequences.
>
> **[Table A] Qauntitative evaluation on temporal scalability.**
>
> |#Frames/PE-res|PMD(↓)|FID(↓)|
> |-|-|-|
> |100/6|0.0010|0.00342|
> |330/6|0.0008|0.00598|
> |660/6|0.0012|0.01251|
> |660/8|0.0010|0.00475|
>
> To support substantially longer sequences, the model can be scaled in two ways: (i) increasing the MLP capacity, or (ii) partitioning the video into temporal segments and optimizing a dedicated motion-field MLP per segment. This segmentation strategy effectively prevents error accumulation and maintains stable optimization over long durations.
>
> **W2. 2D-to-3D ambiguities.** We agree that persistent self-occlusion introduces inherent ambiguity in monocular 2D-to-3D motion transfer. Nevertheless, our approach provides two advantages that help mitigate these effects.
>
> (1) First, our representation is comparably robust to transient occlusion through temporal smoothness regularization from MLP and globally shared morphology parameters. Unlike reconstruct-then-retarget baselines that often suffer from estimation failures when parts are momentarily occluded, our approach ensures that occluded parts remain structurally consistent based on information propagated from other visible frames.
>
> To measure the *degree of view diversity* of monocular videos, we adopt the Effective Multi-view Factor (EMF)[1]:
>
> - Low EMF: near-static viewpoints (severe ambiguity).
> - High EMF: large viewpoint changes give multiple effective views, reducing ambiguity.
>
> We report the results across varying EMF values in Table B. As expected, higher EMF gives the lowest PMD. Importantly, even when the setup approaches a stationary (lowest EMF) configuration, CAMO still maintains stable accuracy and continues to outperform all baselines by a clear margin.
>
> **[Table B] Quantitative evaluations on monocular videos with varying EMF.**
>
> |Camera-viewpoint|EMF(ang)|PMD(↓)|FID(↓)|
> |-|-|-|-|
> |Stationary|35.2|0.0014|0.0075|
> |Slow-orbit(30°)|45.5|0.0011 (NPR:0.0053)|0.0068 (NPR:0.0257)|
> |Teleporting views|253.8|0.0008|0.0083|
>
> (2) Second, our framework naturally incorporates multi-view inputs through differentiable rendering without architectural modifications. This resolves persistent occlusions that cannot be disambiguated from a single view. As shown in Table C and visualized in Appendix A.4.8 (Figure 20), adding even a second view substantially reduces ambiguity.
>
> **[Table C] Quantitative evaluations on multi-veiw motion transfer.**
>
> |#views|PMD(↓)|FID(↓)|
> |-|-|-|
> |1|0.0027 (NPR:0.0045)|0.0145 (NPR:0.0683)|
> |2|0.0022|0.0095|
> |4|0.0020|0.0095|
>
> **W3. Additional references.** We will incorporate both PhysRig [2] and Puppeteer [3] into the Related Work section. We note that PhysRig requires precomputed mesh sequences, and Puppeteer targets ego-character animation, which distinguishes their problem settings from our video-to-3D motion transfer formulation.
>
> **Q1. Regarding ambiguity occurrence** Our analysis shows that most left–right ambiguities disappear under typical casually captured videos, where even modest camera motion (~30° orbit) provides sufficient implicit multi-view cues. Ambiguities primarily arise in degenerate cases when the camera remains nearly fixed at a frontal or profile view and the motion never reveals the occluded limb. In such settings, the 2D projection becomes effectively symmetric, leaving no monocular signal to fully resolve the ambiguity. We provide qualitative examples of this scenario in Figure 17.
>
> ---
>
> *References*
>
> [1] Gao et al., “Monocular Dynamic View Synthesis,” NeurIPS 2022.
>
> [2] Zhang et al., “PhysRig: Differentiable Physics-Based Skinning and Rigging Framework for Realistic Articulated Object Modeling,” arXiv, 2025.
>
> [3] Song et al., “Puppeteer: Rig and Animate Your 3D Models,” arXiv, 2025.

---

> ### Author Response · Authors · 2025-11-26
> **Response to reviewer 2Hny: Update on Related Work**
>
> Dear Reviewer 2Hny,
>
> Following your constructive suggestion (W3), we have carefully integrated the recommended literature into our related work section.
> The updated discussion is located in Section 2 (pages 2-3) of the revised manuscript.
>
> For your convenience, we have also copied the added paragraph below:
>
> *Recent approaches ... explore parametric template-free construction of articulated models .... Specifically targeting character animation, auto-rigging methods[1,2] predict the skeleton and skinning weights of a 3D asset to apply motion extracted from videos or reconstructed mesh sequences. However, these methods typically require a complete morphological[1] or skeletal structural correspondence[2]  between the motion source and the target 3D character.*
>
> ---
>
> **References**
>
> [1] Song et al., “Puppeteer: Rig and Animate Your 3D Models,” arXiv, 2025.
>
> [2] Zhang et al., “PhysRig: Differentiable Physics-Based Skinning and Rigging Framework for Realistic Articulated Object Modeling,” arXiv, 2025.

---

> ### Author Response · Authors · 2025-11-27
>
> Dear reviewer 2Hny,
>
> Thank you again for taking the time to review our work.
>
> We would like to kindly check whether any additional clarification or discussion from our side would be helpful, particularly regarding the points addressed in our recent response.
>
> For your convenience, we summarize the main points and indicate where they are addressed in the revised manuscript below, marked in blue for clarity:
>
> - Temporal scalability (Appendix A.4.3)
> - 2D-to-3D ambiguities (Appendix A.4.7-A.4.8, Figure 17)
> - Include additional references (Sec.2)
>
> Please feel free to let us know if any further details would be useful.
>
> We sincerely appreciate your time and consideration during the review process.

---

### Official Review · Reviewer_JyNN · 2025-11-01

**Soundness:** 3
**Presentation:** 3
**Contribution:** 2
**Rating:** 4
**Confidence:** 4

**Summary:**

This paper presents CAMO, a category-agnostic framework for transferring articulated 3D motion directly from monocular 2D videos to arbitrary 3D target meshes, without relying on category-specific templates or explicit 3D supervision.
The method builds upon articulated 3D Gaussian Splatting (3DGS) and introduces:

Morphology-adaptive parameterization (learnable bone lengths, local Gaussian offsets, and global scaling).

Dense 2D–3D semantic correspondence based on foundation features.

Joint optimization of pose and shape with photometric and semantic losses.

Experiments on Mixamo, DeformingThings4D, and real-world videos show clear improvements over baselines like SPT+, NPR+, and Transfer4D.

**Strengths:**

Category-agnostic capability.
The paper convincingly demonstrates that CAMO generalizes to both humanoids and non-humanoid animals, addressing the typical limitation of category-specific template models.

Clear ablations and metrics.
The quantitative improvements on PMD and FID are substantial (up to 85% improvement on non-quadruped categories). The ablation study effectively supports the importance of morphology parameterization and semantic keypoint supervision.

**Weaknesses:**

**Evaluation is largely self-contained and lacks cross-domain tests.**
While the results on Mixamo and DT4D are consistent, these datasets are synthetic and often aligned in topology.
Real-world results (Fig. 6) are qualitative only, without any perceptual or human study evaluation.
There’s no analysis of failure cases (e.g., severe occlusions, topology mismatch, or multi-actor scenes).

**Limited discussion on theoretical implications and generalization.**
The “morphology-parameterized” model is empirically motivated, but the paper does not analyze why this representation improves optimization stability or disentangles shape/pose effectively.

**No discussion on identifiability or optimization convergence issues under only 2D supervision.**

**Writing and clarity issues.**
The text contains numerous redundancies and long sentences; several sections (e.g., Sec. 3.1–3.3) are nearly copied from AnyMo.
Figures could be more informative — e.g., Fig. 2 is conceptually overloaded but lacks a clear depiction of data flow or loss supervision points.

**Questions:**

How robust is CAMO under non-articulated or highly deformable motions (e.g., cloth, jellyfish, smoke-like motions)?

How is the orientation-sensitive feature extractor trained or selected? Is it frozen or fine-tuned during optimization?

Does the method handle camera motion explicitly, or does it assume static background and known intrinsic/extrinsic parameters?

How does the method scale to longer sequences (e.g., 1K+ frames)? Is optimization stable or prone to drift?

---

> ### Author Response · Authors · 2025-11-22
> **Response to reviewer JyNN (1)**
>
> We thank the reviewer JyNN for the thoughtful and constructive feedback. In response to the reviewer’s valuable suggestions, we address each point in detail below.
>
> **W1. Regarding Evaluation.**
>
> (1) Real-world Evaluation: We acknowledge the reviewer's valid point regarding the reliance on synthetic datasets. To address the lack of rigorous real-world evaluation,
>
> 1. *Qualitative Video Comparison:* We add side-by-side video comparisons with baselines on in-the-wild videos in the supplementary video (00:00~00:30), showing our method’s stability in unconstrained settings.
> 2. *Motion Fidelity & Alignment Metrics:* Although ground-truth 3D poses are unavailable for real-world videos, we will report the M-score (MotionCritic [1]) or temporal consistency score for human subjects by estimating 3D poses from our transferred results.
>
> For broader categories, we note that template-based models (e.g., human or quadruped) are constrained to their predefined domains, whereas our framework is naturally capable of extending to diverse non-template categories (e.g., birds, dolphins).
>
> (2) Regarding challenging or failure case analysis: We expanded our discussion of challenging scenarios in Appendix A.4.3–A.4.4. We include detailed analyses of several representative difficulties: its limitations on long temporal sequences (Figure 15), and the performance drop that occurs under severe or persistent self-occlusion (Figure 17).
>
> **W2. Discussion on theoretical implications and generalization.**
> We thank the reviewer for pointing out the need for deeper theoretical justification. To address this, we have added a new *Theoretical Analysis* section (Appendix B).
>
> In this section, we provide formal proofs demonstrating that our morphology parameterization:
>
> 1. *Alleviates shape-pose ambiguity:* We show that naïve formulations based on freely optimized Gaussian centers lead to inherent shape–pose entanglement under monocular supervision, motivating the adoption of our morphology parameterization.
> 2. *Guarantees identifiability:* Under standard assumptions of piecewise rigidity and non-degenerate motion, our kinematic parameterization guarantees that morphology and pose are uniquely identifiable up to a global scale.
> 3. *Ensures optimization stability:* Our hierarchical parameterization separates global scale, bone lengths, local offsets, and joint rotations, reducing undesirable coupling between parameter groups and leading to more stable optimization.
>
> **W3. Improving writing and clarity.**
> We thank the reviewer for the detailed feedback regarding the presentation. We will refine the manuscript to eliminate redundancies and improve clarity. In particular, we will revise sections 3.1–3.3 to provide a stronger connection to the theoretical background of our morphology-adaptive parameterization, ensuring that the design choices are well-justified by the formal analysis provided in the appendix. We will also redesign Figure 2 to explicitly visualize the end-to-end data flow and clarify the specific supervision points for rendering and keypoint losses.
>
> **Q1. Non-articulated / Highly Deformable Motion.** Thank you for raising this clarification point. CAMO is built upon an articulated skeletal representation (Sec. 3). Consequently, non-rigid dynamics such as cloth, smoke, or fluids are outside our current scope, as they lack the underlying kinematic structure required for our morphology parameterization. Simply applying our approach in such scenarios may not yield reliable results. However, extending our work to integrate physics-based dynamics for 3D Gaussians [2] to handle such soft-body deformations is indeed a compelling direction for future work.
>
> **Q2. Implementational details of the feature extractor.** We employ a pre-trained feature extractor [3]  without fine-tuning during optimization. We validate the effectiveness of this geometry-aware module within our pipeline through ablation studies (Appendix A.3.4, Table 6).
>
> ---
>
> **References**
>
> [1] Wang, Haoru, et al. "Aligning human motion generation with human perceptions." arXiv 2024.
>
> [2] Lin, Yuchen, et al. "OmniphysGS: 3d constitutive gaussians for general physics-based dynamics generation." arXiv 2025.
>
> [3] Zhang, Junyi, et al. "Telling left from right: Identifying geometry-aware semantic correspondence." CVPR. 2024.

---

> ### Author Response · Authors · 2025-11-22
> **Response to reviewer JyNN (2)**
>
> **Q3. Camera Motion and Parameters.** Our framework does not require or estimate the camera pose of the source video. The source sequence is treated purely as a 2D observation stream, and we do not attempt to recover camera trajectory, intrinsics, or extrinsics.
>
> For rendering the target mesh, we adopt a fixed, internally defined perspective camera that establishes the rendering coordinate frame. This camera is not meant to approximate the real source camera; it simply serves as the projection model for differentiable rendering. The 3D pose of the target asset is optimized in this canonical rendering coordinate system to match the source frames in image space (Appendix A.1).
>
> **Q4. Long sequence scalability.** Our method takes the timestamp $t$ as input, applies a sinusoidal positional encoding, and uses a lightweight MLP to parameterize the motion field. With the default temporal frequency, we observed a gradual reduction in fidelity beyond ~600 frames. Increasing the frequency of the sinusoidal embedding mitigates this issue and preserves motion consistency over longer sequences. As shown in Table A, raising the frequency band from 6 to 8 improves performance (PMD 0.0012 → 0.0010, FID 0.01251 → 0.00475), indicating that higher-frequency temporal encoding maintains discriminability over longer sequences.
>
> To support substantially longer sequences, the model can be scaled in two ways: (i) increasing the MLP capacity, or (ii) partitioning the video into temporal segments and optimizing a dedicated motion-field MLP per segment. This segmentation strategy effectively prevents error accumulation and maintains stable optimization over long durations.
>
> **[Table A] Qauntitative evaluation on temporal scalability.**
>
> | # Frames / PE resolution | **PMD ($\downarrow$)** | **FID ($\downarrow$)** |
> | --- | --- | --- |
> | 100 / 6 | 0.0010 | 0.00342 |
> | 330 / 6 | 0.0008 | 0.00598 |
> | 660 / 6 | 0.0012 | 0.01251 |
> | 660 / 8 | 0.0010 | 0.00475 |
>
> We hope our clarifications address the reviewer’s concerns, and we would be happy to answer any further questions.

---

> ### Author Response · Authors · 2025-11-24
> **Response to reviewer JyNN (3)**
>
> Following up on our previous response, we provide the real-world quantitative evaluations and manuscript refinements as detailed below. We are happy to address any additional questions you may have.
>
> **W1. Real-world Evaluation**
>
> To address the concern about the lack of perceptual or human-study evaluations on real-world data, we have conducted additional quantitative perceptual analyses on in-the-wild videos, totaling 6 scenes (humanoids and quadrupeds) with 621 frames.
>
> **Perceptual motion fidelity and alignment metrics.**
>
> For human motion, we evaluate perceptual quality using MotionCritic [1], which is trained to match human perceptual judgments of motion plausibility. As reported in the original paper, critic scores naturally span a wide range (e.g., −18.99 to 20.77 on AMASS[2]), with higher values indicating motions judged as more plausible by humans.
>
> To generalize the evaluation to both human and animal targets, we additionally introduce an LLM-based perceptual score (LLM-as-a-judge) that assesses motion realism, temporal stability, and semantic consistency, following recent multimodal LLM–based video QA approaches [3,4].
>
> We test three multimodal LLM APIs (GPT-4o, GPT-5.1, Gemini-2.5 Flash) using a unified prompt designed to evaluate (i) semantic alignment, (ii) motion quality & physics, and (iii) geometry & mesh integrity. Our evaluation prompt follows the general rubric-based design philosophy of VideoScore [4], but is independently constructed with motion-specific criteria and a 1–10 scoring scale.
>
> **Evaluation results**
>
> The new results (Tables B and C below) show that CAMO consistently outperforms all baselines across both MotionCritic and LLM-based perceptual scores, demonstrating robust performance in unconstrained real-world scenarios.
>
> **[Table B] Quantitative evaluation on humanoid motion transfer**
>
> |  | MotionCritic [1] | GPT-4o | GPT-5.1 | Gemini 2.5 flash |
> | --- | --- | --- | --- | --- |
> | SPT$+$ | -5.33 | 7.67 | 7.50 | 7.33 |
> | Ours | 4.18 | 8.67 | 8.50 | 9.17 |
>
> **[Table C] Quantitative evaluation on quadruped motion transfer**
>
> |  | GPT-4o | GPT-5.1 | Gemini 2.5 flash |
> | --- | --- | --- | --- |
> | NPR$+$ | 7.00 | 8.50 | 4.50 |
> | Ours | 8.83 | 9.00 | 8.67 |
>
> ---
>
> **W3. Improving Writing and Clarity.**
>
> We have revised the manuscript to address the reviewer’s concerns regarding clarity and presentation. In particular:
>
> - Sections **3.1–3.3** have been revised to present the core concepts more directly and improve the overall logical flow.
> - The discussion in **Section 3.2** has been expanded to better motivate the morphology-adaptive parameterization and clarify its theoretical foundations.
> - **Figure 2** has been refined to more clearly illustrate the end-to-end data flow and the supervision signals used by the rendering and keypoint losses.
>
> ---
>
> **References.**
>
> [1] Wang, Haoru, et al. "Aligning Human Motion Generation with Human Perceptions.” ICLR 2025
>
> [2] Mahmood, Naureen, et al. "AMASS: Archive of motion capture as surface shapes.” ICCV, 2019
>
> [3] Q-ALIGN: Teaching LMMs for Visual Scoring via Discrete Text-Defined Levels. ICML, 2024
>
> [4] He, Xuan, et al. "Videoscore: Building automatic metrics to simulate fine-grained human feedback for video generation.” ENMLP, 2024
> ****

---

> ### Author Response · Authors · 2025-11-27
>
> Dear reviewer JyNN,
>
> Thank you again for taking the time to review our work.
>
> We would like to kindly check whether any additional clarification or discussion from our side would be helpful, particularly regarding the points addressed in our recent response.
>
> For your convenience, we summarize the main points and indicate where they are addressed in the revised manuscript below, marked in blue for clarity:
>
> - Qualitative evaluation (supplementary videos)
> - Further analysis on failure case (Figure 17)
> - Writing clarification (Sec. 3.1, 3.2)
> - Updated Figure 2
> - Discussion on theoretical analysis (Appendix B)
> - Scalability to long sequences (Appendix A.4.3, Figure 14, Table 8)
>
> Please feel free to let us know if any further details would be useful.
>
> We sincerely appreciate your time and consideration during the review process.

---

### Author Response · Authors · 2025-11-22
**Common response to all reviewers**

We thank the reviewers for their constructive feedback. In response, we have conducted additional experiments and provided theoretical proofs and visualizations, which are now included in the Appendices and the supplementary video. Below, we provide a categorized summary of these updates and kindly refer the reviewers to the individual responses for detailed discussions.

**1. Supplementary video**

Extended visualizations and qualitative comparisons are presented in the supplementary video. We kindly ask the reviewers to refer to this video for a more comprehensive examination of the qualitative results.
(Anonymized Link: https://drive.google.com/file/d/1tR3ZgRONpi1gE5cwupXrQh7G8957f_2V/view?usp=sharing)

**2. Scalability & Efficiency**

- **Temporal Scalability:** Appendix A.4.3, Figure 14, Table 8 (p.18-19)
- **Multi-view Scalability:** Appendix A.4.8, Table 11, Figure 20 (p.21)
- **Computational Efficiency:** Appendix A.4.5, Figure 18 (p.19-20)

**3. Robustness Analysis**

- **Self-occlusion:** Appendix A.4.7, Table 10 (p.21)
- **Camera Initialization:** Appendix A.4.6, Table 9, Figure 19 (p.20)
- **Fast Motion:** Appendix A.4.9, Table 12 (p.21-22)

**4. Comparisons & Theory**

- **Theoretical Analysis:** Appendix B (p.25-29)
- **Comparison with Generative Pipelines:** Appendix A.6, Figure 21 (p.23-24)
- **Failure Case Analysis:** Figure 17 (p.20)

The revisions can be found in the sections above, where changes are highlighted in blue for clarity. Newly added sections are indicated by coloring the section title in blue.

---

### Author Response · Authors · 2025-12-03

Dear Area Chair,

Thank you for coordinating the review process for our submission. To assist your review process, we provide a 3-minute video summary covering our method and additional experiments. We hope this provides a helpful visual overview.

**anonymized video link (which cannot track visitor info):** https://drive.google.com/file/d/1ZKYYTDYOnj1JecPntrQ7BIxx0PEMjC6B/view?usp=drive_link

---

# Core Contributions

Our work proposes an image-space optimization framework for 2D-to-3D motion retargeting. We briefly summarize the key strengths consistently recognized by the reviewers, who described the work as “structured”, “convincing”, and “likely to stimulate follow-up work”:

- **Category-Agnostic Capability (Highlighted by all):** By eliminating the need for category-specific templates (e.g., SMPL), CAMO works on a wide range of articulated objects (humanoids, quadrupeds, or fantasy characters). All reviewers complemented this versatility as a strength, highlighting it as “a robust, well-implemented, and innovative approach”.
- **Sound & Principled Design:** We bypass the error-prone *“reconstruct-then-retarget”* pipeline using Morphology-Adaptive Articulated 3D Gaussian splatting. Reviewers acknowledged this design as “both elegant and technically solid”(gp2U), “reasonable and coherent” (2Hny), and effective in “reducing error cascade” (S7f1).
- **Scalable representation:** Our pipeline enables efficient 3D motion acquisition from diverse videos, where empirical results clearly validate this, as reviewers acknowledged “substantial improvements on PMD/FID” (JyNN) with “clear empirical gains” (S7f1) over baselines.

# Review Summary

Thankfully, reviewers have provided constructive comments. The table below summarizes each point and where it is addressed in the revised manuscript. The revised sentences and titles of new sections are highlighted in blue.

|Comments|JyNN|2Hny|gp2U|S7f1|Sections|Summary|
|-|-|-|-|-|-|-|
|Temporal scalability|✓|✓|✓||Appendix A.4.3, Tab.8, Fig.14-15|A1.(a)|
|2D-to-3D ambiguity||✓||✓|Appendix A.4.7-A.4.9, Tab.10-12, Fig.20|A1.(b)|
|Robustness on camera initialization||||✓|Appendix A.4.6, Fig.19, Tab.9|A1.(c)|
|Computational efficiency|||✓||Appendix A.4.5, Fig.18|A1.(d)|
|Evaluation on real-world data|✓||||Response to JyNN (3) Tab.B, C|A2.|
|Comparison with generative pipelines||||✓|Appendix A.6., Fig.21|A3.|
|Theoretical Analysis|✓||||Appendix B|A4.|
|Extended visualization||||✓|supplementary video|-|
|*Rating*|4|6|6|4|||

# Detailed Response Summary

|A1.|Robustness and scalability|
|-|-|
|(a)|**Temporal scalability:** We demonstrated that single MLP scales up to 600-frame sequences by increasing time embedding frequency. We further discussed a temporal split strategy as a scalable solution for longer videos.|
|(b)|**2D-to-3D ambiguity:** To address concerns about ambiguity under self-occlusion and fast motion/blur, we showed stability through low-EMF analysis for self-occlusion and robust handling of fast motion. Additionally, multi-view experiments demonstrate how our method effectively scales to exploit multi-view cues.|
|(c)|**Robustness on Camera initialization:** Our method show robustness under extreme camera pose perturbations (up to 180°), thanks to our hierarchical optimization strategy.|
|(d)|**Computational Efficiency:** We demonstrate consistent optimization times (~5 mins) across varying frame lengths and mesh resolutions.|
|**A2.**|**Evaluation on Real-World Data**|
||We included quantitative real-world evaluations via perceptual metrics and scene-by-scene video comparisons. Evaluations using MotionCritic and LLM scoring demonstrate CAMO's superiority over baselines across diverse in-the-wild sequences.|
|**A3.**|**Comparison with Generative Pipelines**|
||We included a comparison against a generative pipeline (Gen-4 followed by Stag4D). The results indicate that CAMO provides superior identity preservation and more faithful motion retargeting, while generative approaches often exhibit identity and motion drift.|
|**A4.**|**Theoretical Analysis on Morphology Parameterization**|
||We added a detailed Theoretical Analysis (Appendix B), which provides conditions under which our morphology-parameterized Gaussian articulation (1) mitigates shape-pose entanglement, (2) yields identifiability of shape and pose parameters up to a global scale, and (3) explains the improved optimization stability.|

**Additional revisions in the manuscript**

- Presentation (JyNN): We revised Sec. 3 and updated Fig. 2 to improve clarity regarding the core concepts and data flow with supervision signals.
- Citations (2Hny): We updated Sec. 2 to include the missing references suggested by the reviewer.
- Future Work (JyNN, gp2U): We expanded the discussion on non-articulated motions and physical realism.

We hope the revisions offer clearer explanations of our contributions, and we thank you for your thoughtful consideration.

Sincerely, The Authors

---

### Meta-Review · Area_Chair_YYRF · 2026-01-10

**Summary:**

The paper proposes CAMO, a category-agnostic 2D-to-3D motion transfer framework leveraging morphology-adaptive 3D Gaussian splatting and dense 2D–3D semantic correspondences. While technically sound and well-presented, reviewers highlighted limitations in robustness, temporal scalability, and evaluation. Real-world experiments remain limited, with heavy occlusion, fast motion, and topology mismatches only partially addressed. Theoretical insights are mostly empirical, and physical realism or convergence analyses are incomplete. Overall, the contributions are incremental relative to existing 3D motion-transfer and differentiable rendering methods.

**Reviewer Concerns:**

The rebuttal addressed several points: extended real-world quantitative evaluation, long-sequence temporal analysis, and additional visualizations. It clarified feature extractor training and camera initialization robustness. However, fundamental limitations remain: persistent 2D-to-3D ambiguities, reliance on rigging quality, lack of physical motion metrics, and limited analysis under severe occlusions or highly deformable objects. The theoretical discussion, though expanded, does not fully justify generalization, leaving key methodological concerns unresolved.

**Reviewer Scores:**

If reviewers could have updated their scores post-rebuttal, minor improvements might occur due to additional experiments and clarifications. Reviewers JyNN and S7f1 might remain at marginally below acceptance (4), while 2Hny and gp2U might remain marginally above (6), reflecting partial resolution of concerns. Overall, no reviewer appears fully convinced of the method’s robustness, generality, or practical impact, supporting a decision to reject.

---

### Decision · Program_Chairs · 2026-01-26

Reject